# In-parameter Knowledge Injection: Integrating Temporary Contextual Information into Model Parameters

## Abstract

Large Language Models (LLMs) have achieved remarkable performance in various natural language processing tasks by leveraging relevant external knowledge provided by the users or retrieved from external sources. Traditionally, this external information is incorporated by appending it directly to the model's input context, a paradigm known as in-context knowledge injection. However, this paradigm faces significant limitations due to the finite input context length of LLMs and often results in shallow integration between the external knowledge and the model's internal representations. To address the limitations of in-context knowledge injection, we propose a new knowledge injection paradigm called in-parameter knowledge injection, which temporarily embeds the external knowledge relevant to the user's input directly into the model's parameters rather than its input context. This new paradigm overcomes the context length limitations of LLMs and enables deeper integration of external information within the model's internal parametric representations. Through extensive experiments across tasks of varying complexity, we demonstrate that in-parameter knowledge injection achieves significant benefits for complex tasks requiring intricate reasoning. In contrast, in-context injection remains effective for simpler tasks where answers can be directly extracted from the provided information[1].

## 1 Introduction

Large language models (LLMs) have achieved remarkable success in the field of natural language processing (NLP), demonstrating exceptional capabilities across a variety of tasks (Brown et al., 2020; Chowdhery et al., 2022; Touvron et al., 2023; Scao et al., 2022; Zhang et al., 2022). A critical factor contributing to their success is their ability to utilize external knowledge effectively and efficiently, thereby improving their performance on specific tasks (Lewis et al., 2020). In practical applications, this external knowledge typically consists of passages either provided directly by users or retrieved from external databases. Given the pivotal role of external knowledge in boosting the performance of LLMs, it becomes imperative to explore effective methods for integrating this information into the models, leading to a crucial research question: How can we effectively integrate external knowledge into large language models to ensure they fully comprehend and internalize the injected information?

Currently, the in-context knowledge injection paradigm is the predominant approach for integrating external knowledge, where relevant information is appended directly to the model's input (Dong et al., 2022; Lewis et al., 2020; Levy et al., 2024). This method is widely adopted due to its simplicity in implementation. However, it has notable limitations. The finite length of the LLMs' input context restricts the amount of external knowledge that can be incorporated (Levy et al., 2024). More importantly, as shown by previous studies, language models process knowledge in input prompts and model parameters differently(Nanda et al., 2023), which means that simply adding information to the input is not enough to activate the full power of language models' knowledge reasoning abilities. These limitations become especially pronounced in tasks that require multi-hop inference or advanced reasoning over the injected knowledge (Li et al., 2024; Levy et al., 2024).

---

[1]We have open-sourced all the code, data, and models in the following anonymous GitHub link: https://anonymous.4open.science/r/In-parameter-Knowledge-Injection/

In light of these limitations, we investigate a critical yet underexplored research question: ***Is there a more effective paradigm to integrate external knowledge (e.g., a few passages) into LLMs than simply appending it to their input?***

To address this research question, we propose a new knowledge injection paradigm called in-parameter knowledge injection (PKI), which is parallel to the in-context knowledge injection (CKI) paradigm. Unlike the in-context paradigm that appends information to the input, our proposed in-parameter paradigm embeds external knowledge directly into the model's parameters. This approach overcomes the input-length limitations of in-context methods and allows for deeper integration of external information within the model's internal parametric knowledge representations.

Under the in-parameter knowledge injection paradigm, numerous methods can be developed to integrate external knowledge into the parameter of language models. As the first work to propose this paradigm, we present a simple and effective method to highlight its potential and encourage further exploration in this direction. Our method begins by expanding each external passage $p$ into a set of passage-question-answer tuples. Leveraging this augmented dataset, we apply parameter-efficient fine-tuning (PEFT) methods such as Adapter (Houlsby et al., 2019) and LoRA (Hu et al., 2022) to inject the passage into additional parameters. This process allows us to temporarily integrate the external knowledge for a specific query without permanently modifying the original parameters.

We conduct a series of experiments that progressively increase task complexity and reasoning depth to evaluate the effectiveness of our proposed in-parameter knowledge injection paradigm compared to the traditional in-context paradigm. Our findings reveal a clear trend: as the complexity and the depth of reasoning increase, methods under the in-parameter paradigm demonstrate superior performance. To be specific, for tasks demanding advanced reasoning, such as multi-document reading comprehension and multi-hop inference across multiple documents, our proposed in-parameter knowledge injection paradigm significantly outperforms the in-context paradigm. Conversely, the in-context approach remains more effective for straightforward tasks requiring direct answer extraction from the provided passage. These results underscore the importance of aligning the knowledge injection strategy with the specific demands of the task.

In conclusion, the contributions of this paper are as follows:

- We propose a new knowledge injection paradigm for LLMs, i.e., in-parameter knowledge injection, which directly embeds external passages into the parameters of LLMs, addressing the limitations of the in-context knowledge injection paradigm.

- Under this new paradigm, we introduce a simple yet effective method utilizing data augmentation and parameter-efficient fine-tuning techniques to effectively integrate external knowledge into LLMs' parameters.

- We conduct extensive experiments across various scenarios to compare the performance of in-context and in-parameter knowledge injection methods. Our findings highlight the conditions under which each approach is most effective, providing practical guidance on selecting the most suitable method based on task requirements.

## 2 PROBLEM FORMULATION OF KNOWLEDGE INJECTION

Knowledge injection for language models involves temporarily integrating relevant external information to aid the model in executing a specific query. The relevant external information (e.g., a few passages or documents) is utilized solely for this query and is subsequently discarded. The main goal of knowledge injection is to enhance the model's performance by incorporating the selected relevant information that is not adequately covered in its pre-training data. Two primary paradigms for injecting this knowledge into language models are in-context and in-parameter, as illustrated in Figure 1. In this section, we provide a formal definition of these two paradigms, explaining their distinct mechanisms for injecting knowledge.

### 2.1 IN-CONTEXT KNOWLEDGE INJECTION PARADIGM

In the in-context knowledge injection paradigm, the relevant passages are appended directly to the model's input as part of the prompt. This paradigm injects the external information into prompt

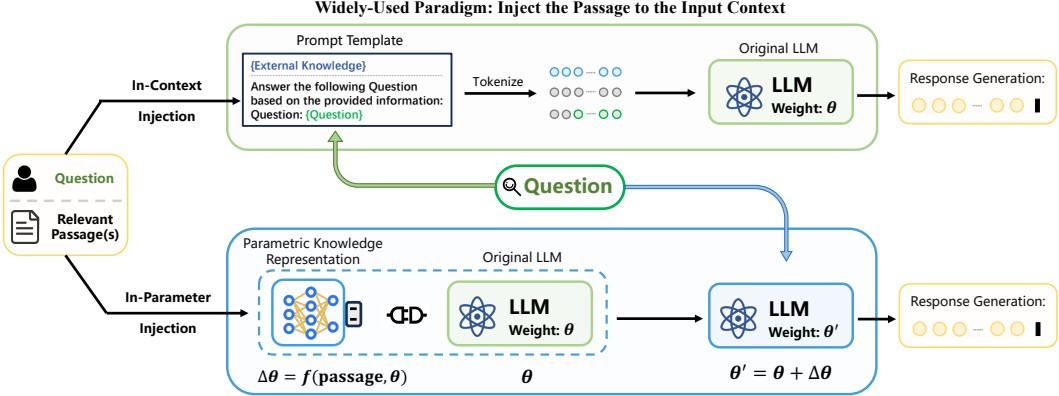

Figure 1: An illustration of the comparison of in-context and in-parameter knowledge augmentation paradigms: In-context injection combines relevant passages and the query in the input, using the original LLM $\theta$ to answer the question without modifying its parameters. Our proposed In-parameter injection temporarily updates the LLM's parameters $\theta' = \theta + \Delta\theta$ specifically for the given user query, temporarily integrating relevant knowledge into the parameter for this question.

templates which guide the model to generate a response based on the provided information. For a given task, we have the query $q$ and the external passages $\mathcal{K}$. The input to the model becomes $(q, \mathcal{K})$, and the model is instructed to generate the answer $a$ based on $q$ and $\mathcal{K}$.

The advantage of the in-context paradigm is that it provides a simple and direct way to inject knowledge into the model, making it straightforward to implement. However, this method faces limitations due to the finite input context length of LLMs and results in shallow integration between the injected knowledge and the model's internal representations.

## 2.2 IN-PARAMETER KNOWLEDGE INJECTION PARADIGM

In this paper, we explore a new paradigm for integrating the selected passages into LLMs than the conventional method of appending them to the input. We propose in-parameter knowledge injection (PKI), a paradigm that embeds external knowledge directly into the model's parameters instead of the context. To formalize the PKI paradigm, we define a parameter update function $f_\phi$ parameterized by $\phi$. This function takes the external passages $\mathcal{K}$ and the current model parameters $\theta$ as inputs to compute a conditional parameter shift $\Delta\theta$. The computation is formalized as:

$$\Delta\theta = f_\phi(\mathcal{K}, \theta), \ \ \theta' = \theta + \Delta\theta, \tag{1}$$

where $\Delta\theta$ is the parametric knowledge representation that represents the adjustment to the model parameters necessary to incorporate the external passages. The updated parameters $\theta'$ integrate this knowledge directly into the model. The updated model $\mathcal{M}'$ with parameters $\theta'$ is then used to generate the answer $a$.

Unlike knowledge editing methods such as Knowledge Neurons (Dai et al., 2021), Rank-One Model Editing (Meng et al., 2022), and Self-Edit (Liu et al., 2024), which focus on permanently modifying a specific piece of knowledge in a model by identifying and altering certain neurons, our PKI paradigm temporarily integrates knowledge from entire passages into the model's parameters to address specific queries. This allows for quick, query-specific knowledge updates without permanently changing the model, much like appending the passage to the context but without the limitations of context length or the need for repeated processing. Our approach also differs from continued pre-training, which retrains the LLM on the entire knowledge base, significantly altering its parameters and requiring substantial time and resources. In contrast, PKI allows for quick, query-specific knowledge updates without permanently altering the model's underlying knowledge.

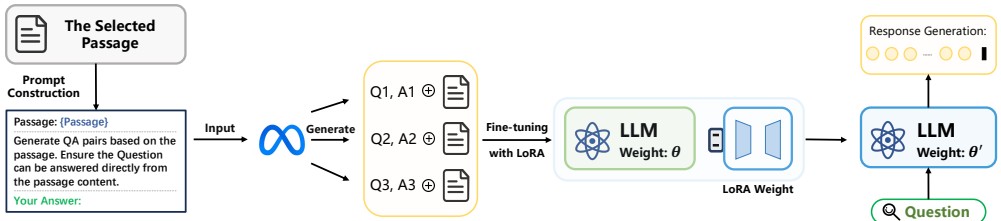

Figure 2: An illustration of our proposed in-parameter knowledge injection framework. The prompt shown is a simplified representation. The detailed prompt is in Appendix A.

## 3 METHODOLOGY OF IN-PARAMETER KNOWLEDGE INJECTION

Our proposed In-Parameter Knowledge Injection method, illustrated in Figure 2, integrates the passage $\mathcal{K}$ directly into the parameters of a pre-trained language model $\mathcal{M}$ by computing parametric knowledge representations that can be directly applied to the model's parameters. The entire paradigm consists of two key phases: the *parametric knowledge encoding* phase and the *generation* phase. In the following sections, we will provide a detailed explanation of each phase.

### 3.1 PARAMETRIC KNOWLEDGE ENCODING PHASE

As described in Section 2.2, given a question $q$ and relevant knowledge $k$, we aim to integrate $k$ into the language model's parameter and enable the language model to utilize this knowledge for subsequent tasks effectively. The purpose of the Parametric Knowledge Encoding (PKE) Phase is precisely designed to achieve this. In the PKE Phase, we encode the external knowledge $\mathcal{K}$ into parametric knowledge representation that can be directly injected into the model's parameters $\theta$. The process consists of two steps: Data Augmentation and Parametric Knowledge Integration, which are detailed below.

**Data Augmentation**    To effectively inject the **external knowledge** $\mathcal{K}$ into the language model, we first augment each passage $k \in \mathcal{K}$ into a suitable format for knowledge integration. This process involves generating question-answer (QA) pairs for each passage, which serve as the basis for training the language model to integrate the knowledge. Specifically, we employ a language model, which can be either an auxiliary or the primary model (denoted as $\mathcal{A}$), to generate QA pairs from each passage. For each $k \in \mathcal{K}$, we apply a structured prompt template $\mathcal{T}$ to format the passage before passing it to $\mathcal{A}$. This prompt template is carefully designed to elicit informative and diverse QA pairs that reflect the core content of the passage (detailed in Appendix A). Using the prompt template $\mathcal{T}$, the language model $\mathcal{A}$ generates a set of QA pairs for each passage $k$. We aggregate all generated QA pairs into a dataset $\mathcal{D}$:

$$\mathcal{D} = \bigcup_{k \in \mathcal{K}} \{(k, u_i, a_i) \mid i \in 1, 2, \ldots, n\}, \qquad (2)$$

where $\mathcal{D}$ represents the collection of all tuples $(k, u_i, a_i)$, each $(u_i, a_i)$ pair corresponds to a specific question and answer derived from passage $k$, and $n$ is the number of QA pairs we generated, which is a tunable hyperparameter.

**Parametric Knowledge Integration**    To incorporate the knowledge from the selected passages into the language model, we introduce additional parameters that represent this information and integrate them into the original model. Specifically, we employ low-rank adaptations to the model's weight matrices for these extra parameters, which allows us to efficiently adjust the model without the need for full fine-tuning [2]. We initialize these additional parameters and then train them using the augmented dataset $\mathcal{D}$ defined in Equation 2, enabling the model to effectively internalize the new knowledge.

---

[2]Our primary focus is on the framework that enables the model to internalize knowledge from passages. Other methods like Adapter or prefix-tuning could also be utilized to calculate the parametric knowledge representation, which we leave for future work.

Specifically, for each sample $(k, u_i, a_i)$ in $\mathcal{D}$, we first construct the input by concatenating the passage $k$, the question $u_i$, and the answer $a_i$ into a single sequence. Then, we use the standard language modeling objective to train the LLM to predict all the tokens in the input sequence based on all preceding tokens. For each weight matrix $W \in \mathbb{R}^{d \times k}$ in the model parameters $\theta$, we introduce low-rank matrices $A$ and $B$ such that:

$$W' = W + \Delta W = W + AB^\top, \tag{3}$$

where $A \in \mathbb{R}^{d \times r}$, $B \in \mathbb{R}^{k \times r}$, and $r \ll \min(d, k)$. These low-rank matrices $\Delta \theta = \{A, B\}$ constitute the parametric knowledge representation that can be directly integrated into the original model $\mathcal{M}$. The optimization objective is to minimize the negative log-likelihood of the target tokens over the entire input sequence:

$$\min_{\Delta \theta} \mathcal{L}(\theta + \Delta \theta) = - \sum_{(k, u_i, a_i) \in \mathcal{D}} \sum_{t=1}^{T} \log P_{\theta + \Delta \theta}(x_t \mid x_{<t}), \tag{4}$$

where $x = [k; u_i; a_i]$ is the concatenated input sequence, and $T$ is the total number of tokens in the sequence. As gradients are calculated over the entire input, including the passage, the question, and the answer, thereby facilitating the effective knowledge internalization of the entire passage. This comprehensive training strategy ensures that even if certain details are omitted from the question-answer pairs, the passage itself, through repeated exposure during training, becomes embedded within the model's parameters.

The resulting additional parameters $\Delta \theta$ serve as the parametric knowledge representation, which can be directly added to the original model $\mathcal{M}$ to enhance its performance on tasks requiring the integrated knowledge. Importantly, this entire process can be performed offline; each passage or a group of passages can be processed in advance to compute their respective parametric representations, eliminating the need for real-time computation during online deployment.

## 3.2 GENERATION PHASE

In the Generation Phase, we augment the original model $\mathcal{M}$ with the parametric knowledge representation $\Delta \theta$ to create an updated model $\mathcal{M}'$ with parameters $\theta' = \theta + \Delta \theta$. This updated model is used to generate answers to the question $q$ without providing the knowledge passages $\mathcal{K}$ as input. By internalizing the knowledge into its parameters, the model leverages this information flexibly, enabling deeper reasoning and more informed responses.

## 3.3 EFFICIENCY COMPARISON BETWEEN IN-CONTEXT AND IN-PARAMETER PARADIGM

The in-context paradigm appends external knowledge directly to the input prompt during inference, increasing the input length and the computational resources required. Specifically, it demands more GPU memory due to the longer sequences processed by the model's self-attention mechanism, whose computational complexity scales quadratically with input length. The in-parameter paradigm involves encoding the parametric knowledge representation. This process consumes time and memory for calculating the representation, **but it can be performed offline**. As a result, the model can simply load these pre-computed parameters for real-time queries without additional processing overhead. When combining IP and IC (IP+IC), the additional computational cost introduced by IP is negligible compared to the cost of IC alone. The majority of the computational overhead comes from processing the longer input sequences in IC, while IP's additional FLOPs are minimal and can be performed offline.

In summary, the effectiveness of the in-parameter method alone may not always match that of the in-context method across all tasks. Combining IP and IC can leverage the strengths of both approaches and in such cases, the additional computational overhead introduced by IP is minimal. The in-parameter method front-loads the computational effort during an offline training phase and thus benefits from reduced inference time and memory usage, making it capable of handling real-time queries.

## 4 EXPERIMENTAL SETUP

### 4.1 TASKS AND DATASETS

We design a series of progressive experiments tailored to incrementally increase both difficulty and the requisite amount of reasoning needed for the resolution. This structured approach allows us to evaluate the effectiveness of different knowledge injection methods under varying complexities. To be specific, our experiments span from simple fact extraction to complex multi-hop reasoning tasks, using a variety of datasets suited to each task's requirements.

**Level 1: Fact Extraction Task** Our initial experiments focus on queries that ask about explicit facts directly present in the relevant passage or document without requiring additional reasoning. This is the simplest form of query, where the model's primary task is to locate and extract the relevant information. For this experiment, we use the TriviaQA dataset (Joshi et al., 2017), which consists of question-answer pairs where the answers are explicit facts found within the given passages.

**Level 2: Comparative Reasoning Task** To introduce a higher level of complexity, we consider tasks that require simple reasoning over information extracted from two documents. Specifically, we focus on **comparison questions**, which involve comparing two or more entities from the same group based on certain attributes. For example, a question might ask, *"Who was born first, Bill Clinton or Donald Trump?"* Answering such questions requires the model to extract relevant facts from multiple documents and perform a comparison. We utilize the **Comparison** subset of the 2WikiMultihopQA (2WQA) dataset (Ho et al., 2020) for this task.

**Level 3: Multi-Step Comparative Reasoning Task** Further increasing the difficulty, we examine **bridge-comparison questions**, which require an additional reasoning step for each extracted answer before performing the comparison. For instance, instead of directly comparing two books, a question might ask, *"Which book has the author born first, Pride and Prejudice or 1984?"* To answer, the model needs to identify the authors of the books and then compare their dates of birth. We use the **Bridge Comparison** subset of the 2WQA dataset for this task.

### 4.2 EVALUATION METRICS

For all tasks, we evaluate the models based on their ability to provide correct answers. We extract the final answer from the generated output using pattern-matching techniques. The extracted answer is then compared with the reference answer, utilizing methods such as exact match at the answer level, along with token-level measurements of the F1 score. The details of our experimental settings, including the instructions provided to the models and the implementation of evaluation are provided in Appendix B.

### 4.3 IMPLEMENTATION DETAILS

For in-context knowledge injection, we directly concatenate the relevant passages to the prompt template and input the combined text into the language model's context. The specific prompt templates and the designing process are detailed in Appendix D. For in-parameter knowledge injection, we employ GPT-4o as the external model to generate question-answer (QA) pairs based on the passages. The specific configurations, including batch size, epochs, learning rate, and LoRA parameters, are detailed in Appendix B. The experiments utilized the Qwen2.5-1.5B-Instruct (Yang et al., 2024) and LLaMA3.2-1B-Instruct (Meta, 2024) and LLaMA-3-8B models, with all conducted using PyTorch on 40GB NVIDIA A100 GPUs. The generation settings, including the decoding strategy, and hardware specifics, are detailed in Appendix B.

## 5 EXPERIMENTAL RESULTS

In this section, we comprehensively evaluate different knowledge injection methods across tasks with varying levels of complexity. Our objective is to understand how each method performs under different reasoning demands and to identify the circumstances under which each method is most effective. The methods compared are:

Table 1: Comparison of In-context and In-Parameter knowledge injection methods for the Fact Extraction Task, with the best results in bold and second-best results underlined. An accompanying diagram on the left illustrates the task's complexity.

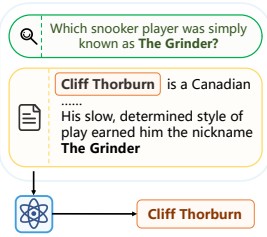

|  | Qwen-1.5B | | LLaMA-1B | | LLaMA-8B | |
| --- | --- | --- | --- | --- | --- | --- |
| Method | EM | F1 | EM | F1 | EM | F1 |
| In-Context | 0.4913 | 0.6246 | 0.5896 | **0.6979** | **0.6532** | **0.7787** |
| IC-QA | **0.5202** | 0.6200 | **0.6012** | 0.6962 | 0.6358 | 0.7770 |
| In-Parameter | 0.2486 | 0.3685 | 0.3006 | 0.3903 | 0.4509 | 0.5336 |
| IP & IC | 0.5087 | **0.6604** | 0.5260 | 0.6487 | 0.6350 | 0.7709 |

Table 2: Experimental results of In-context and In-Parameter knowledge injection method on the Comparative Reasoning Task. The best results are in bold and the second-best results are underlined. The diagram on the left illustrates the task difficulty.

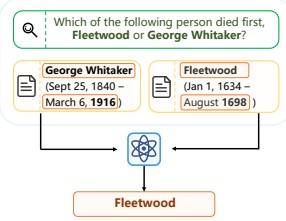

|  | Qwen-1.5B | | LLaMA-1B | | LLaMA-8B | |
| --- | --- | --- | --- | --- | --- | --- |
| Method | EM | F1 | EM | F1 | EM | F1 |
| In-Context | 0.2148 | 0.3240 | 0.2517 | 0.2635 | 0.3767 | 0.4915 |
| IC-QA | 0.1846 | 0.2426 | 0.1342 | 0.1394 | 0.3833 | 0.5576 |
| In-Parameter | 0.3188 | 0.3826 | **0.4765** | **0.5154** | 0.4033 | 0.5793 |
| IP & IC | **0.3960** | **0.4473** | 0.4732 | 0.5127 | **0.5933** | **0.7285** |

- **In-Context**: The traditional In-context knowledge injection method that directly adds the question and relevant passage into the input context of the language model.

- **IC-QA**: Inputting the Passage, Question, and the QA pairs (the same QA pairs generated for the in-parameter method) into the language model. The inclusion of QA pairs in this baseline is designed to ensure a fair comparison with the In-parameter Knowledge Injection method.

- **In-Parameter (IP)**: Injecting knowledge directly into the model's parameters using the method from Section 3.

- **IP & IC**: Combining the In-Parameter method with In-context by both injecting knowledge into the parameters and concatenating the Passage into the input.

By progressively increasing the difficulty level of the tasks—from simple fact extraction to complex multi-step reasoning—we aim to reveal how each knowledge injection method scales with task complexity and reasoning requirements.

## 5.1 FACT EXTRACTION TASK

In this initial task, we assess the models' abilities to extract explicit facts directly present in the provided passages without additional reasoning. This task represents the simplest scenario, focusing on straightforward entity extraction. As shown in Table 1, the **In-context** and **IC-QA** methods outperform the **In-Parameter** methods on both the Qwen and LLaMA models. This outcome suggests that when the required information is readily available in the input, providing the passage directly to the model is the most effective approach.

The superior performance of the in-context methods can be attributed to the models' proficiency in understanding and extracting information from the immediate context. On the other hand, the **In-Parameter** methods underperform in this task. One possible explanation is that injecting knowledge into the parameters may introduce unnecessary complexity for simple tasks and might not capture the precise details or could potentially obscure the exact information needed for fact extraction. These findings highlight that for tasks involving direct extraction of information with minimal reasoning, in-context methods are more advantageous.

Table 3: Performance comparison of In-context and In-Parameter knowledge injection methods on the Multi-Step Comparative Reasoning Task. The best results are in bold and the second-best results are underlined. The diagram on the left illustrates the task difficulty.

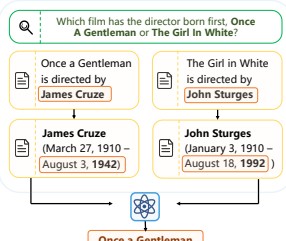

| Method | Qwen-1.5B | | LLaMA-1B | | LLaMA-8B | |
|---|---|---|---|---|---|---|
| | EM | F1 | EM | F1 | EM | F1 |
| **In-Context** | 0.2617 | 0.3578 | 0.3188 | 0.3483 | 0.4333 | 0.6414 |
| **IC-QA** | 0.2181 | 0.2545 | 0.2785 | 0.2969 | 0.3500 | 0.5864 |
| **In-Parameter** | **0.3557** | **0.3964** | 0.3758 | 0.4102 | 0.4652 | 0.6537 |
| **IP & IC** | 0.3289 | 0.3493 | **0.4027** | **0.4278** | **0.6239** | **0.6722** |

## 5.2 COMPARATIVE REASONING TASK

In the next level of complexity, models are required to perform reasoning over information extracted from two documents. This involves comparison questions where the answer is not explicitly stated but must be inferred through basic reasoning. Table 2 reveals a notable shift in performance: the **In-Parameter** and **IP & IC** methods now outperform the in-context approaches on both models. This suggests that as the task requires the integration of multiple pieces of information and reasoning over them, embedding knowledge into the model's parameters becomes more effective.

These findings demonstrate that as complexity and reasoning demands grow, embedding knowledge directly into the model's parameters becomes increasingly beneficial. The In-Parameter method allows the model to internalize external knowledge, enabling deeper integration and more flexible reasoning across multiple pieces of information. Moreover, combining In-Parameter with In-Context knowledge further enhances performance, indicating that parameter-injected knowledge supplemented by contextual information yields better outcomes.

## 5.3 MULTI-STEP COMPARATIVE REASONING TASK

At the highest level of complexity, the models tackle multi-step comparative reasoning tasks that require chaining several inference steps before arriving at the answer. This involves not only extracting information but also performing sequential reasoning over that information. As shown in Table 3, the **In-Parameter** and **IP & IC** methods continue to outperform the in-context methods, with the performance gap widening compared to the previous task. This trend underscores the increasing efficacy of in-parameter knowledge injection as the reasoning demands escalate.

A possible reason for this is that the in-parameter methods enable the models to handle the complexity of multi-hop reasoning more effectively. By embedding knowledge into the parameters, the models develop richer internal representations that support complex inferential chains. This internalization allows for more sophisticated reasoning. The combined **IP & IC** method often yields the best performance, suggesting a synergistic effect. Providing the model with both internalized knowledge and contextual information may facilitate reasoning by offering multiple avenues for accessing and processing the necessary data. In contrast, the in-context methods face some challenges in this task. The necessity to conduct multiple reasoning steps within the input context likely overwhelms the models, leading to diminished performance. This highlights the limitations of relying solely on in-context information for complex reasoning tasks.

## 5.4 ANALYSIS

The experiments indicate a clear trend: in-parameter knowledge injection becomes more effective as task complexity and reasoning demands increase, enhancing the model's deeper reasoning capabilities. In-context methods perform better in simple fact extraction tasks as models can readily utilize the provided context. However, for higher-order reasoning, in-context approaches are limited by the capacity to reason based on the injected knowledge. In such cases, in-parameter methods provide an advantage by embedding knowledge internally, enabling deeper reasoning based on its parametric knowledge. These findings suggest that selecting a knowledge injection method based on the task complexity can significantly enhance model performance.

Table 4: Comparison of performance with and without few-shot chain-of-thought (CoT) prompting. The best and second-best results are highlighted in bold and underlined, respectively. The left figure illustrates the setting. The models used are Qwen2.5-1.5B-Instruct and LLaMA3.2-1B-Instruct.

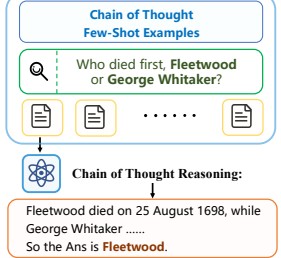

| Method | Comparative Reasoning | | Multi-Setp Comparison | |
|---|---|---|---|---|
| | Qwen | LLaMA | Qwen | LLaMA |
| **In-Context** | 0.3154 | 0.3960 | 0.4094 | 0.2013 |
| **IC-QA** | 0.2987 | 0.3020 | **0.4195** | 0.2483 |
| **In-Parameter** | 0.3456 | 0.4497 | 0.4027 | **0.4094** |
| **IP & IC** | **0.3859** | **0.4765** | 0.3993 | 0.3523 |

# 6 ABLATION STUDY

## 6.1 INFLUENCE OF FEW-SHOT CHAIN-OF-THOUGHT EXAMPLES

To evaluate whether incorporating few-shot chain-of-thought (CoT) (Wei et al., 2023) examples affects our conclusions, we introduced CoT into the prompts for the Comparative Reasoning Task and the Multi-Step Comparative Reasoning Task. The experimental results are shown in Table 4.

The In-Parameter and IP & IC methods continued to outperform the in-context approaches. This consistency suggests that embedding knowledge directly into the model's parameters is inherently effective for complex reasoning tasks, regardless of the presence of CoT examples in the prompts. These findings indicate that our earlier conclusions about the superiority of in-parameter knowledge injection are robust and not significantly influenced by the addition of CoT examples. The models' ability to handle intricate reasoning appears to rely more on internalized knowledge than on explicit reasoning cues provided during inference.

## 6.2 EFFECT OF QA PAIR GENERATION

To evaluate the effectiveness of the QA pair generation step in our In-Parameter Knowledge Injection method, we conducted an ablation study comparing two training strategies:

- Passage-Only Training: The language model is fine-tuned directly on the knowledge passages $\mathcal{K}$ using the standard language modeling objective, without generating QA pairs.
- Passage with Generated QA Training: Our proposed method, where an auxiliary language model generates QA pairs from each passage, and the model is fine-tuned on the concatenated sequences of passage, question, and answer.

As shown in Table 5, incorporating generated QA pairs into the training process significantly enhances the model's performance. The model trained with QA pairs outperforms the passage-only model by a substantial margin across all evaluation metrics, demonstrating the effectiveness of our approach. The notable performance improvement can be attributed to the explicit question-answering context provided by the generated QA pairs. Training on these pairs enables the model to better internalize and organize the external knowledge, learning not just the memorize the content of the passages but also how to apply this knowledge to respond to specific queries. In contrast, passage-only training lacks this targeted learning mechanism, leading to less effective knowledge integration and application.

# 7 RELATED WORKS

**In-Context Knowledge Injection**   In-context knowledge injection is a prevalent method for augmenting language models with external information by appending relevant passages directly to the input context. This approach has been widely used in tasks such as reading comprehension and Retrieval-Augmented Generation (RAG), where models generate responses based on both the

Table 5: Ablation study results comparing Passage-Only Training and Passage with Generated QA Training. The best results are in bold.

| | | Qwen2.5-1.5B-Instruct | | LLaMA3.2-1B-Instruct | |
|---|---|---|---|---|---|
| Method | QA | EM | F1 | EM | F1 |
| In-Parameter | w QA | **0.3188** | **0.3826** | **0.4765** | **0.5154** |
| | w/o QA | 0.0168 | 0.0773 | 0.3926 | 0.4246 |
| IP & IC | w QA | **0.3960** | **0.4473** | **0.4732** | **0.5127** |
| | w/o QA | 0.1107 | 0.1926 | 0.2819 | 0.3123 |

prompt and the injected knowledge (Lewis et al., 2020; Zhou et al., 2024; Su et al., 2024). RAG systems typically retrieve relevant documents and incorporate them into the input to enhance performance on specific tasks. To improve the efficacy of in-context knowledge injection, some studies have focused on optimizing prompts and instructions. For example, Trivedi et al. (2022) introduced IR-CoT, which investigates how to design prompts and few-shot examples to effectively integrate knowledge into the context, thereby enhancing the model's reasoning capabilities over the injected information. To address the limited context window of language models, various context compression techniques have been proposed to mitigate constraints on the amount of external knowledge that can be included (Ge et al., 2023; Verma, 2024).

**In-Parameter Knowledge Injection** In contrast, in-parameter knowledge injection embeds external knowledge directly into the model's parameters, offering the potential to incorporate more extensive and nuanced information without the constraints of input length. This approach is relatively underexplored, with the most closely related areas being knowledge editing and continued pre-training. Knowledge editing methods, such as Knowledge Neurons (Dai et al., 2021), Rank-One Model Editing (Meng et al., 2022), and Self-Edit (Liu et al., 2024), permanently modify language models to incorporate new information, typically addressing specific facts or entities. These approaches are not designed for the temporary integration of extensive external knowledge tailored to specific tasks. Continued pre-training increases a model's knowledge through further training on extra data, yet it requires substantial resources and time. To alleviate this, parameter-efficient fine-tuning (PEFT) techniques like LoRA (Hu et al., 2022), Adapter (Houlsby et al., 2019), and Prefix-Tuning (Li & Liang, 2021) serve as alternatives. These methods allow efficient integration of knowledge with minimal updates to model parameters. For instance, LoRA adjusts model weights with low-rank updates, and Prefix-Tuning enhances input sequences with a learnable prefix. **Although these approaches improve task performance efficiently, they mainly focus on task adaptation rather than quick, query-specific knowledge updates.**

In contrast, our method temporarily injects external knowledge directly into the model's parameters by encoding the knowledge into parametric knowledge representation. Much like directly appending a passage to the input context, our method does not permanently alter the parameters of the LLM. Instead, the passage related to a specific query is temporarily embedded into the parameters, serving only that query.

## 8 CONCLUSION

This paper introduces a novel knowledge injection paradigm, in-parameter knowledge injection, as an alternative to the traditional in-context knowledge injection approach. By directly integrating external knowledge into the parameters of generative language models through representation learning, this paradigm overcomes several limitations associated with the in-context approach. Specifically, it eliminates the dependency on extensive prompt engineering, reduces the need for large context windows, and enhances the efficiency and adaptability of language models in handling knowledge-intensive tasks. Our experiments compared in-context and in-parameter methods across diverse tasks. Results show that in-parameter knowledge injection excels in reasoning-intensive scenarios, such as advanced comprehension and multi-document inference, due to deeper knowledge integration. In contrast, the in-context method is better suited for simple tasks where answers are directly extractable. The selection between these two methods in practical applications depends on a trade-off: in-parameter offers performance gains for complex reasoning, while in-context minimizes latency for simpler tasks. Practitioners should decide based on specific application needs.

## 9 REPRODUCIBILITY STATEMENT

To ensure the reproducibility of our results, we have thoroughly documented all settings and details within the paper. Moreover, we have meticulously organized and open-sourced all the code, data, and models used in this study. These resources are available at the following anonymous GitHub link: `https://anonymous.4open.science/r/In-parameter-Knowledge-Injection/`.

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

## A    PROMPT TEMPLATE FOR QA GENERATION

In order to systematically generate high-quality question-answer (QA) pairs from each knowledge passage, we designed a specialized prompt template for the auxiliary language model. The primary objective of this prompt is to ensure that the generated QA pairs are both informative and diverse, accurately reflecting the essential content of the original passage. This structured approach facilitates effective knowledge integration by providing the language model with clear and consistent training data.

### A.1    DESIGN CONSIDERATIONS

Several key factors influenced the design of the prompt template:

- **Clarity and Specificity**: The prompt explicitly instructs the model to generate three distinct questions, each answerable using the passage.

- **Structured Output Format**: By specifying the exact format for the questions and answers, including fields such as "question", "answer", and "full_answer", we facilitate automated parsing and processing of the generated data. We used a simple example to help the model understand the format we need.

- **Quality Control**: Requiring that each question be answerable using the provided passage helps maintain the relevance and accuracy of the QA pairs, preventing the introduction of extraneous or speculative information. This clarity helps in minimizing ambiguity and ensures that the generated QA pairs are directly relevant to the source material.

### A.2    PROMPT TEMPLATE

The following prompt template is used to guide the auxiliary language model in generating QA pairs:

---

**Prompt Template for QA Generation**

I will provide a passage of text, and you need to generate three different questions based on the content of this passage. Each question should be answerable using the information provided in the passage. Additionally, please provide an appropriate answer for each question derived from the passage.
You need to generate the question and answer in the following format:
```
[
    {
        "question": "What is the capital of France?",
        "answer": "Paris",
        "full_answer": "The capital of France is Paris."
    },
]
```
This list should have at least 3 elements. You only need to output this list in the above format.
Passage:
{passage}

---

When applying the prompt template, the passage is dynamically inserted into the designated placeholder, ensuring that each QA generation task is contextually tied to its corresponding knowledge excerpt. The auxiliary language model processes this prompt to produce a structured list of simple single-hop QA pairs, adhering strictly to the specified format.

## B    EXPERIMENT DETAILS

This appendix provides a comprehensive overview of the models used, the experimental setting, and the evaluation methodologies employed in our study.

### B.1 MODEL DESCRIPTIONS

In our experiments, we utilized two language models Qwen2.5-1.5B-Instruct and LLaMA3.2-1B-Instruct. Thes instruction-tuned models excel in dialogue applications, offering enhanced performance for complex tasks.

- **Qwen2.5-1.5B-Instruct** (Yang et al., 2024): Qwen2.5-1.5B-Instruct is part of the latest Qwen2.5 series of multilingual large language models, featuring 1.5 billion parameters. This instruction-tuned, text-only model is optimized for multilingual dialogue use cases, including agentic retrieval and summarization tasks. Qwen2.5-1.5B-Instruct supports long-context handling of up to 128K tokens and is fine-tuned using supervised fine-tuning (SFT) and reinforcement learning with human feedback (RLHF) to enhance helpfulness and safety.
- **LLaMA3.2-1B-Instruct** (Meta, 2024): LLaMA3.2-1B-Instruct model is an instruction-tuned, text-only variant optimized for multilingual dialogue use cases, including agentic retrieval and summarization tasks. Architecturally, LLaMA3.2 employs an optimized transformer-based auto-regressive framework, ensuring efficient and scalable language generation capabilities. The instruction-tuned versions leverage supervised fine-tuning (SFT) and reinforcement learning with human feedback (RLHF) to better align with human preferences for helpfulness and safety.

### B.2 EXPERIMENTAL SETTING

Our experimental framework comprises two primary datasets: TriviaQA and 2WikiMultihopQA. Below, we provide detailed descriptions of the setting and procedures applied to each.

#### B.2.1 2WIKIMULTIHOPQA

We employed two data types from the 2WikiMultihopQA dataset: Comparison and Bridge Comparison. For each data type, the first 298 questions were selected to generate responses for evaluation during the main experiment. The ablation study experiment for LLaMA3-8B-Instruct utilizes the first 103 questions.

For every question within these categories, we extracted the pertinent information and created three question-answer pairs for each extracted segment. To enhance the model's ability to recall and effectively utilize passage information, we organized the training data for in-parameter training as follows:

- **(Q, P, A) Prompts**: Two QA pairs combine the question (Q), passage (P), and answer (A), encouraging the model to associate the passage with the corresponding answer.
- **(Q, A) Prompts**: One QA pair includes only the question and answer, facilitating the model's ability to generate answers based on the learned passage content.

#### B.2.2 TRIVIAQA

We employed the Wikipedia development set from TriviaQA, where all relevant documents are sourced from Wikipedia. To reduce the complexity of reading comprehension for the model, we implemented a filtering process that retained only passages containing 3,000 tokens or fewer for each question. Initially, the dataset included 308 questions; after filtering, 173 questions remained, each containing at least one relevant piece of information within the length limit. These filtered questions were used for our experiments.

For every question within these categories, we extracted the pertinent information and created six question-answer pairs for each extracted segment. Following the aforementioned format in 2WikiMultihopQA, each question includes three (Q, P, A) prompts and three (Q, A) prompts to enhance the model's ability to effectively recall and utilize passage information.

### B.3 EVALUATION DETAILS

To assess the performance of our models, we employed two primary evaluation metrics: Exact Match (EM) and F1 Score.

For tasks that require direct answer generation, we incorporate the prompt phrase "The answer is" within the input. This encourages the model to produce the answer immediately following this phrase, ensuring consistency with the expected output format.

For tasks that involve reasoning and necessitate the generation of a Chain-of-Thought (CoT), we provide eight few-shot examples that utilize CoT. Each few-shot example concludes with the phrase "So, the answer is xxx", which guides the model to follow this pattern and facilitates accurate answer matching.

### B.4 TRAINING PARAMETER CONFIGURATION

The training parameters and LoRA parameters used in our In-Parameter Knowledge Injection method are on Table 6.

Table 6: Configuration of Training Parameters

| Parameter | Value |
|---|---|
| Number of Training Epochs | 3 |
| Learning Rate | $3 \times 10^{-4}$ |
| LoRA Alpha | 32 |
| LoRA Dropout | 0.01 |
| LoRA Rank | Qwen2.5-1.5B-Instruct: 2
LLaMA3.2-1B-Instruct: 2
LLaMA3-8B-Instruct: 16 |

## C   MORE ABLATION STUDY RESULTS

### C.1   EFFECT OF LoRA PARAMETERS

To investigate the impact of LoRA parameters, we conducted an ablation study using the LLaMA3.2-1B-Instruct model on the first 103 questions of the Comparison dataset. We systematically varied the rank, $\alpha$ and dropout rates, as detailed in Table 7, to assess their effects on performance.

The experimental results indicate that changes in LoRA parameters have little influence on the model's performance. For smaller models, a lower rank typically leads to better performance. The alpha parameter has a relatively minor impact, while a lower dropout rate also contributes to improved performance.

Table 7: Ablation study results comparing different rank, $\alpha$, and dropout configurations. The best results are in bold.

| Rank | $\alpha$ | Dropout | EM | F1 | Prec. | Recall |
|---|---|---|---|---|---|---|
| 2 | 16 | 0.01 | 0.4854 | 0.4999 | 0.4977 | 0.5100 |
| 2 | 32 | 0.01 | **0.5631** | **0.5994** | **0.5959** | **0.6081** |
| 2 | 32 | 0.05 | 0.4175 | 0.4244 | 0.4236 | 0.4291 |
| 4 | 32 | 0.01 | 0.5243 | 0.5267 | 0.5275 | 0.5262 |
| 8 | 32 | 0.01 | 0.4563 | 0.4636 | 0.4628 | 0.4680 |
| 16 | 32 | 0.01 | 0.4466 | 0.4555 | 0.4547 | 0.4583 |

### C.2   EFFECT OF THE NUMBER OF TRAINING EPOCHS AND KNOWLEDGE-AUGMENTED QAS

To investigate the influence of training parameters on model performance, we conducted an ablation study using the LLaMA3.2-1B-Instruct model on the first 103 questions of the Comparison dataset. Specifically, we examined the effects of varying the number of training epochs (#Epoch) and the number of knowledge-augmented question-answer pairs (#QA). The performance is presented in Tables 8 and 9.

Table 8: Ablation study results comparing different numbers of training epochs. The best results are in bold.

| #Epoch | EM | F1 | Prec. | Recall |
|--------|--------|--------|--------|--------|
| 1 | 0.5534 | 0.5733 | 0.5744 | 0.5748 |
| 3 | **0.5631** | **0.5994** | **0.5959** | **0.6081** |
| 5 | 0.3981 | 0.4099 | 0.4091 | 0.4146 |
| 7 | 0.3592 | 0.3893 | 0.3859 | 0.4008 |

The results in Table 8 indicate that three training epochs yield the best performance across all metrics. Training for a single epoch also achieves relatively strong results, but increasing the number of epochs beyond three leads to a significant decline in performance. Similarly, Table 9 shows that increasing the number of knowledge-augmented QAs up to three substantially improves the model's performance, with the optimal results observed at three QAs. However, adding more than three QAs results in decreased performance.

These results suggest that excessive training may lead to overfitting or weaken the model's ability to generalize effectively, ultimately causing it to lose its ability to answer questions accurately.

Table 9: Ablation study results comparing different numbers of knowledge-augmented QA. The best results are in bold.

| #QA | EM | F1 | Prec. | Recall |
|-----|--------|--------|--------|--------|
| 1 | 0.4272 | 0.4387 | 0.4384 | 0.4453 |
| 2 | 0.4757 | 0.4911 | 0.4896 | 0.4947 |
| 3 | **0.5631** | **0.5994** | **0.5959** | **0.6081** |
| 4 | 0.4369 | 0.4523 | 0.4502 | 0.4615 |
| 5 | 0.4272 | 0.4535 | 0.4503 | 0.4647 |
| 6 | 0.4175 | 0.4199 | 0.4207 | 0.4194 |

## D  PROMPT DETAILS

For methods that perform In-Context Learning solely on passages, we utilize the following prompt:

---

**Prompt Template for In-Context Learning with Passages Only**

user:

You should answer the question by referring to the knowledge provided below and integrating your own knowledge.
Passage 1: Blind Shaft is a 2003 film about a pair of brutal con artists operating in the illegal coal mines of present- day northern China. The film was written and directed by Li Yang, and is based on Chinese writer Liu Qingbang's short novel" Shen MuSacred Wood").
Passage 2: The Mask of Fu Manchu is a 1932 pre-Code adventure film directed by Charles Brabin. It was written by Irene Kuhn, Edgar Allan Woolf and John Willard based on the 1932 novel of the same name by Sax Rohmer. Starring Boris Karloff as Fu Manchu, and featuring Myrna Loy as his depraved daughter, the movie revolves around Fu Manchu's quest for the golden sword and mask of Genghis Khan. Lewis Stone plays his nemesis. Dr. Petrie is absent from this film.

Question: Which film came out first, Blind Shaft or The Mask Of Fu Manchu?

assistant:

The answer is

---

For the (Q, P, A) part of the In-Parameter training data, the data is the prompt above with the answer added at the end.

For the part that does not use knowledge, we use the following prompt:

---

**Prompt Template for In-Context Learning with Passages Only**

user:

You should answer the question by referring to the knowledge provided below and integrating your own knowledge.

Question: Which film came out first, Blind Shaft or The Mask Of Fu Manchu?

assistant:

The answer is

---

For the (Q, A) part of the In-Parameter training data, the data is the prompt above with the answer added at the end.

For methods that perform In-Context Learning on both passages and QA-generated knowledge, we use the following prompt:

---

**Prompt Template for In-Context Learning with Passages and Knowledge-Augmented QA**

user:

You should answer the question by referring to the knowledge provided below and integrating your own knowledge.
Passage 1: Blind Shaft is a 2003 film about a pair of brutal con artists operating in the illegal coal mines of present-day northern China. The film was written and directed by Li Yang, and is based on Chinese writer Liu Qingbang's short novel "Shen Mu (Sacred Wood)".
Passage 2: The Mask of Fu Manchu is a 1932 pre-Code adventure film directed by Charles Brabin. It was written by Irene Kuhn, Edgar Allan Woolf, and John Willard based on the 1932 novel of the same name by Sax Rohmer. Starring Boris Karloff as Fu Manchu, and featuring Myrna Loy as his depraved daughter, the movie revolves around Fu Manchu's quest for the golden sword and mask of Genghis Khan. Lewis Stone plays his nemesis. Dr. Petrie is absent from this film.

Here are some questions and answers about the knowledge.
Question: What is the film "Blind Shaft" about?
Answer: A pair of brutal con artists operating in the illegal coal mines of present-day northern China.
Question: Who wrote and directed the film "Blind Shaft"?
Answer: Li Yang.
Question: What is the source material for the film "Blind Shaft"?
Answer: Chinese writer Liu Qingbang's short novel "Shen Mu (Sacred Wood)".
Question: Who directed the film The Mask of Fu Manchu?
Answer: Charles Brabin.
Question: Who played the character Fu Manchu in the film?
Answer: Boris Karloff.
Question: What is Fu Manchu seeking in the movie?
Answer: The golden sword and mask of Genghis Khan.

You need to answer the question: Which film came out first, Blind Shaft or The Mask of Fu Manchu?

assistant:

The answer is

---

For tasks that require guiding the model to generate Chain-of-Thought (CoT) reasoning, we use the following instruction to provide few-shot examples before the external knowledge:

---

**Prompt Template for CoT fewshot**

You should reference the knowledge provided below and combine it with your own knowledge to answer the question. Please follow the format of the example I provided above.
Here are some examples about how to answer the questions.
Question: When did the director of film Hypocrite (Film) die?
Answer: The film Hypocrite was directed by Miguel Morayta. Miguel Morayta died on 19 June 2013. So the answer is 19 June 2013.

Question: Are both Kurram Garhi and Trojkrsti located in the same country?
Answer: Kurram Garhi is located in the country of Pakistan. Trojkrsti is located in the country of Republic of Macedonia. Thus, they are not in the same country. So the answer is no.

Question: Do director of film Coolie No. 1 (1995 Film) and director of film The Sensational Trial have the same nationality?
Answer: Coolie No. 1 (1995 film) was directed by David Dhawan. The Sensational Trial was directed by Karl Freund. David Dhawan's nationality is India. Karl Freund's nationality is Germany. Thus, they do not have the same nationality. So the answer is no.

Question: Who is Boraqchin (Wife Of Ögedei)'s father-in-law?
Answer: Boraqchin is married to Ögedei Khan. Ögedei Khan's father is Genghis Khan. Thus, Boraqchin's father-in-law is Genghis Khan. So the answer is Genghis Khan.

Question: Who was born first out of Martin Hodge and Ivania Martinich?
Answer: Martin Hodge was born on 4 February 1959. Ivania Martinich was born on 25 July 1995. Thus, Martin Hodge was born first. So the answer is Martin Hodge.

Question: When did the director of film Laughter In Hell die?
Answer: The film Laughter In Hell was directed by Edward L. Cahn. Edward L. Cahn died on August 25, 1963. So the answer is August 25, 1963.

Question: Which film has the director died later, The Gal Who Took the West or Twenty Plus Two?
Answer: The film Twenty Plus Two was directed by Joseph M. Newman. The Gal Who Took the West was directed by Frederick de Cordova. Joseph M. Newman died on January 23, 2006. Fred de Cordova died on September 15, 2001. Thus, the person to die later from the two is Twenty Plus Two. So the answer is Twenty Plus Two.

Question: Who is the grandchild of Krishna Shah (Nepalese Royal)?
Answer: Krishna Shah has a child named Rudra Shah. Rudra Shah has a child named Prithvipati Shah. Thus, Krishna Shah has a grandchild named Prithvipati Shah. So the answer is Prithvipati Shah.

---

The passages and knowledge-augmented question-answer pairs follow the same format as above. In the generation part, we guide the model to think step-by-step.

---

**Prompt Template for CoT generation**

user:

......

Let's think step by step. Answer the questions in the same format as above.

Question: Which film came out first, Blind Shaft or The Mask Of Fu Manchu?

assistant:

Answer:

---

