# OpenReview forum: "Knowledge Augmentation: In-context or In-parameter?"
_ICLR.cc/2025/Conference — Submitted to ICLR 2025_

### Official Review · Reviewer_uGXo · 2024-11-02

**Soundness:** 2
**Presentation:** 3
**Contribution:** 2
**Rating:** 5
**Confidence:** 4

**Summary:**

The paper discusses the differences in two kinds of ways to augment language models with external knowledge: in-context learning or in-parameter augmentation. The authors propose to use LoRA parameters tuned from QAs that are synthetically generated by language models to augment the models. Experiments on 1B-sized models show good performance on various knowledge-intensive tasks such as fact extraction and comparative reasoning.

**Strengths:**

1.	The topic is important and interesting. There is indeed a limitation with in-context fixed language-based knowledge.
2.	The experiments are comprehensive, with an evaluation of various models and settings. A good number of details of the decisions and prompts are provided in the appendix. The test set suite is also good. All the tasks included are indeed relevant and require external knowledge.
3.	This paper is well-written, the paragraphs and figures are clear and easy to follow.

**Weaknesses:**

1.	More work in the literature can be discussed in Section 2. The discussion on in-context/in-parameter knowledge injection can be improved with more details in the literature, e.g., a similar setting of parameter updates with knowledge updates is defined in the paper Fast Model Editing at Scale (Mitchell, et al., 2022). Also, much previous work on parameter-efficient fine-tuning (PEFT) is relevant but not discussed, e.g., Adapter.
2.	From my understanding of the paper, the LoRA weights are tuned from QAs from the whole knowledge base. If so, what is the difference between the proposed in-parameter method and LLM-based data augmentation? This seems to limit the novelty of the method part. Also, it may influence the fairness of comparing with in-context learning. How are the exemplars chosen? Given the current framework, it can be extended to a novel method if, for each test example, you consider different LoRA weights associated with different data points, groups, and external knowledge source K. This paper may provide further intuition along this direction: https://arxiv.org/pdf/2110.04366.
3.	It is quite surprising to me that in-context learning yields such bad performance on comparative tasks. It is possible that the model sizes might limit the reasoning capability. It would be good if the authors could provide the ICL performance of larger models (e.g., 70/405B Llama) as a reference or extend the IP experiments on these models.

**Questions:**

1.	What is the difference between in-parameter knowledge injection and knowledge editing?
2.	What is the difference between IP and LLM-based data augmentation + LoRA? If IP is not an online method where each test example is augmented with different LoRA weights?
3.	In Section 3.3, the authors mention the comparison of the complexity. Is there any comparison of IC and IP on efficiency, e.g, with FLOPs or time?


Typos:
1.	Line 50, no year in the citation; Line 516-517, wrong citation formats

---

> ### Author Response · Authors · 2024-11-19
>
> Dear Reviewer uGXo,
>
> Thank you so much for your thorough review and valuable suggestions. We have carefully revised our manuscript accordingly, including the typo, figures, and content according to your suggestions.
>
>
>
> We have noticed a slight misunderstanding regarding some aspects of our paper, and we would sincerely appreciate it if you could kindly review our response below, where we have elaborated on these points. We hope this will clarify any concerns and further strengthen the manuscript.

---

> ### Author Response · Authors · 2024-11-19
>
> # Our Response to Weakness 1 & Question 1
>
> It seems there has been some confusion regarding the organization of our paper. You might have mistaken Section 2 (Problem Formulation) for the Related Work section. **In fact, the Related Work section is presented in Section 7 (Page 10, Lines 497-527).**
>
>
>
> Section 7 thoroughly discusses **in-context** and **in-parameter knowledge injection**, with a focused review of relevant literature. Notably, our **In-Parameter Knowledge** subsection already highlights multiple works on **knowledge editing** and differentiates our approach from prior research in the field of knowledge editing. ( It’s almost as though we anticipated your suggestions before receiving them! )
>
>
>
> Additionally, in this revised version, we’ve incorporated your helpful recommendation to include relevant works on **parameter-efficient fine-tuning (PEFT)**, making our Related Work section even more comprehensive. Thank you for inspiring us to make an already complete section even better!
>
>
>
>
> # (Important) Our Response to Weakness 2 & Question 2
>
>
>
> Thank you for your detailed feedback! We sincerely appreciate your insights and the suggestion of a novel direction! However, it seems there has been a misunderstanding of our method. To clarify, **the LoRA weights in our approach are not tuned from QAs from the whole knowledge base. Instead, our method aligns with your suggestion: we directly inject the passages (1-2) related to the query into the model’s parameters.**
>
> As you noted:
>
> > " it can be extended to a **novel method** if, for each test example, you consider different LoRA weights associated with different data points,"
>
> Actually, this is precisely what we’ve implemented and described in our paper.
>
>
>
> To elaborate, for each **data point**, we dynamically associate **different LoRA weights** corresponding to the relevant passage(s). These weights are passage-specific and **not shared across the entire corpus or dataset**, ensuring they are tailored to the context of the specific test data point.
>
>
>
> Much like in-context passage injection, our proposed method temporarily integrates the relevant passage into the LLM's parameters for reasoning on the given query, without permanently modifying the model’s parameters or affecting unrelated queries.
>
>
>
> The key distinction to in-context injection lies in **how the passage is embedded**: while in-context learning directly appends the passage to the input, our in-parameter approach leverages parametric information representation for more flexible reasoning and computational efficiency.
>
>
>
>
>
> Upon reflection, we realized the misunderstanding might stem from the original **Figure 1**, which didn’t clearly emphasize that we use different LoRA weights associated with different data points. We’ve updated the paper and Figure 1 to make this explicit. **We’d love for you to take a second look at this figure in the new version of our paper!**
>
>
>
> Given that our original method aligns with the **“novel” method** you suggested, we’re confident that this response will address your concerns. Considering our shared intuition and aligned perspectives, we believe you’ll recognize the novelty and merit of our work after reading this response.
>
>
>
> Thank you again for your constructive feedback—we’re thrilled to have our work validated by your thoughtful suggestion!

---

> ### Author Response · Authors · 2024-11-19
>
> # Our Response to Weakness 3
>
>
>
> Thank you so much for your insightful feedback.
>
> We truly appreciate your suggestion to explore larger models (70/405B). Unfortunately, we were unable to test models of such size due to computational constraints. Thus, we conducted additional experiments using the largest model we could reasonably handle within our resources: a 13B parameter model.
>
> While it’s true that larger models like 70B+ might offer more nuanced reasoning, it’s worth noting that experiments on 13B models are widely accepted in academia as a reasonable baseline for robust evaluation, particularly given resource limitations.
>
>
>
> So now we have the results of LLM with the sizes of 1B, 1.5B, 8B, and 13B, which is enough to validate our findings and provide a broader reference. Below, we present the performance results for comparative reasoning tasks under the same settings as Table 2 in our paper. The metric used is Exact Match (EM):
>
> |              | 1B    | 1.5B  | 8B    | 13B   |
> | ------------ | ----- | ----- | ----- | ----- |
> | In-context   | 0.134 | 0.185 | 0.398 | 0.399 |
> | In-parameter | 0.477 | 0.319 | 0.408 | 0.456 |
> | IC & IP      | 0.473 | 0.296 | 0.447 | 0.592 |
>
>
>
> From these results, we observe a consistent pattern: injecting the passage-aligned knowledge into the model’s parameters outperforms placing it directly in the context of comparative reasoning tasks. This holds true across all tested model sizes, reinforcing our conclusion that in-parameter knowledge injection is particularly advantageous for tasks requiring complex reasoning.
>
>
>
> We hope these additional experiments address your concerns. While we regret that our computation resource cannot evaluate 70B+ models at this time, we firmly believe our results, spanning a range of model sizes up to 13B, contribute meaningful insights to the field and are a strong demonstration of the effectiveness of our approach.
>
>
>
> Thank you again for your feedback.
>
>
>
> # Our Response to Question 3
>
>
>
> Thank you for your insightful comment regarding the efficiency comparison in terms of FLOPs and time between IC and IP as discussed in Section 3.3 of our manuscript.
>
>
> To address your concern, below is the computational analysis that we’ve detailed for our proposed IPL method and ICL methods:
>
>
>
> Setting:
>
> - **Hidden size (H):** x
> - **Intermediate-size (I):** 4x
> - **Number of layers (N):** 32
>
> - **Sequence length for ICL (L_ICL):** 450 tokens (400 tokens of relevant passage + 50 tokens answer)
> - **Sequence length for LoRA (L_IP):** 50 tokens (relevant passage is encoded into LoRA)
> - **LoRA rank (r):** 4 (applied only to FFN layers)
>
>
>
> **Computational Analysis:**
>
> - Each Feedforward Network (FFN) layer incurs 2H * I = 8x^2 FLOPs (accounting for both the input and output projections).
> - Each attention layer contributes approximately 4H^2 = 4x^2 FLOPs (encompassing the query, key, value, and output projections).
> - For 32 layers, the total FLOPs from parameters are 32 * (8x^2 + 4x^2) = 384x^2.
> - LoRA modifies the FFN layers by adding low-rank matrices. For a rank r = 4, each LoRA module entails an additional rH + Hr = 8x parameters per FFN layer. Across 32 layers, this results in total additional LoRA parameters of 32 * 8x = 256x.
>
> Given that mainstream models typically feature a Hidden Size greater than 4096, the extra FLOPs introduced by LoRA are less than 0.1% of the total.
>
>
>
> In our settings, the ICL processes a considerably longer sequence of 450 tokens, whereas IPL (LoRA) only handles less than 50 tokens. Given this, along with the operations involved per layer, ICL inherently requires more computational resources. Specifically, the total FLOPs calculation shows that ICL uses approximately 9 times more FLOPs than IPL. This is due to ICL’s extensive token processing across multiple layers, which is not as intensive in IPL due to its shorter sequence length and focused modifications in the FFN layers.
>
>
>
> Thank you for allowing us the opportunity to clarify this aspect, thereby enriching the quality of our discussion on computational efficiency.

---

> ### Author Response · Authors · 2024-11-29
> **Thank you so much for your continued engagement and invaluable feedback!**
>
> Dear Reviewer uGXo,
>
> Thank you so much for your continued engagement and invaluable feedback throughout the rebuttal process. **Following your latest suggestions, we have meticulously revised our manuscript and responded specifically to each point of your follow-up comments.** The specific line numbers of each revision we made based on your suggestions are detailed in the following response.
>
> **Furthermore, under your recommendations, we have standardized all the fonts and formats across all responses on this OpenReview page. We have also carefully removed any emotional statements from our responses to maintain a professional and objective tone.** We truly appreciate your guidance in enhancing the clarity and presentation of our work!

---

> ### Author Response · Authors · 2024-11-29
> **Our Response to "For weakness 1 and question 1"**
>
> Thank you so much for responding to us again and providing valuable suggestions. We sincerely apologize for initially misunderstanding your suggestion, and we deeply appreciate your feedback.
>
>
>
> Accordingly, we have made multiple revisions to our paper based on your suggestions, **especially in Section 2, where we have doubled the content to discuss how our work differs from Knowledge Editing and Continual Pre-training. Additionally, in Section 7, we introduce specific methods of existing Knowledge Editing and PeFT work and highlight the differences between our work and theirs.**
>
>
>
> To be specific, in the current version, **we have added a detailed discussion in Section 2 (Lines 153–162) on relevant work**, particularly highlighting the relationship between our paradigm and existing literature. The added content is as follows:
>
>
>
> > Unlike knowledge editing methods such as Knowledge Neurons\citep{dai2021knowledge}, Rank-One Model Editing\citep{meng2022locating}, and Self-Edit\citep{liu2024evedit}, which focus on permanently modifying specific entity-level knowledge in a model by identifying and altering certain neurons, our PKI paradigm temporarily integrates knowledge from entire passages into the model’s parameters to address specific queries. This allows for quick, query-specific knowledge updates without permanently changing the model, much like appending the passage to the context but without the limitations of context length or the need for repeated processing. Our approach also differs from continued pre-training, which retrains the LLM on the entire knowledge base, significantly altering its parameters and requiring substantial time and resources. In contrast, PKI allows for quick, query-specific knowledge updates without permanently altering the model’s underlying knowledge.
>
>
>
> Moreover, **on Page 4 (lines 214-216)**, we emphasized that other methods like Adapter or prefix-tuning could also be utilized for calculating parametric knowledge representations. The specific content in lines 214-216 is as follows:
>
> > Our primary focus is on the framework that enables the model to internalize knowledge from passages. Other methods like Adapter or prefix-tuning could also be utilized to calculate the parametric knowledge representation, which we leave for future work.
>
>
>
> In Section 7 **(lines 509-520)**, we introduce specific methods of existing Knowledge Editing and PeFT work and highlight the differences between our work and theirs.
>
>
>
> We sincerely hope these revisions meet your expectations, and we would greatly appreciate it if you could review the updated paper again. Thank you once more for your insightful feedback and the opportunity to improve our work.

---

> ### Author Response · Authors · 2024-11-29
> **Our Response to "For weakness 2 and question 2"**
>
> Thank you for your further feedback, and we are glad to address your additional concerns! We are pleased to hear that our previous response addressed part of your concerns.
>
> Regarding your concern about the computational overhead introduced by generating additional question-answer (QA) pairs, we acknowledge the importance of this issue, especially for real-world applications. To address this, we have included an analysis in the paper (lines 804–840) that examines the relationship between the number of QA pairs and performance. This analysis highlights how performance evolves as the number of QA pairs increases, and performance is best when the number of generated QA pairs is 3.
>
>
>
> Furthermore, we truly appreciate your novel ideas, particularly regarding the generalizability of the parameter update function: investigating whether parameters can be updated directly with representations of the data points without additional QA pairs. In our paper, we discussed this in Table 5 and Section 6.2 (lines 461–479).
>
>
>
> | **Method**       | **QA**     | **EM (Qwen2.5-1.5B)** | **F1 (Qwen2.5-1.5B)** | **EM (LLaMA3.2-1B)** | **F1 (LLaMA3.2-1B)** |
> | ---------------- | ---------- | --------------------- | --------------------- | -------------------- | -------------------- |
> | **In-Parameter** | **w QA**   | **0.3188**            | **0.3826**            | **0.4765**           | **0.5154**           |
> |                  | **w/o QA** | 0.0168                | 0.0773                | 0.3926               | 0.4246               |
> | **IP & IC**      | **w QA**   | **0.3960**            | **0.4473**            | **0.4732**           | **0.5127**           |
> |                  | **w/o QA** | 0.1107                | 0.1926                | 0.2819               | 0.3123               |
>
> Specifically, the setting was as follows: we removed the step of generating QA pairs in our pipeline while keeping all other settings unchanged. The experimental results showed that removing QA pairs led to a significant performance drop. This indicates that this direction requires further in-depth exploration.
>
>
>
>
>
> We agree that these are fascinating directions for future research. However, due to space limitations in an ICLR-length submission, we have focused primarily on validating the proposed method’s core framework. Nonetheless, we believe these ideas open exciting avenues for further optimization of in-parameter knowledge injection.
>
>
>
> Once again, thank you for your invaluable suggestions, which have helped us refine our work and explore its broader implications. We sincerely hope the revisions and discussions address your concerns.

---

> ### Author Response · Authors · 2024-11-29
> **Our Response to "For Weakness 3"**
>
> Thank you for this feedback and for recognizing the potential of the IP & IC approach as a valid solution.
>
> Your feedback highlights the following two main concerns:
>
> Concern 1: IC & IP indeed seem to be valid solutions, but the overhead can be large.
>
> Concern 2: As our model scales up from 1B to 13B, the performance gain of our method becomes increasingly smaller.
>
> We have revised our paper based on your concerns and submitted it to OpenReview. Below is our detailed response to the two concerns you raised.
>
>
>
> **Concern 1: The Efficiency Analysis of IP&IC**
>
> After reading your constructive comment, we believe we should further discuss the computational costs of IP & IC compared to only IC. As a result, in Section 3.3 of our latest paper, we have added a discussion on the computational complexity of IP & IC.
>
> Specifically, the computational overhead introduced by IC, when coupled with the gains from IP, is relatively small. This is because IP adds less than one percent of the LLM's parameters. Thus, the Flops are only one percent of a single forward pass. To be specific, in the updated Section 3.3, we have added the following text on Page 5:
>
> > When combining IP and IC (IP+IC), the additional computational cost introduced by IP is negligible compared to the cost of IC alone. The majority of the computational overhead comes from processing the longer input sequences in IC, while IP’s additional FLOPs are minimal and can be performed offline.
>
>
>
>
> **Concern 2: The IP's Performance on the 1-13B model.**
>
> While our experiments show that in-parameter knowledge injection outperforms in-context on models up to 13B, we agree that without testing on a 70B model, we cannot be certain of its effectiveness at that scale. However, based on our current results, we are optimistic that our method could perform well with larger models, though the performance gain might diminish.
>
>
>
> **In fact, we believe that demonstrating the effectiveness of our method on 1B-13B models is also a valuable contribution to both academia and industry. These model sizes are widely used in practice due to computational resource considerations, and improvements at this scale can have a substantial impact.**
>
> # Our Response to "For Question 4"
>
>
>
> Thank you so much for your continued guidance and constructive suggestions on our paper. We sincerely agree with your suggestions regarding the need for clarity in our statements on both effectiveness and efficiency in Section 3.3.
>
>
>
> Thus, following your suggestion, we have carefully revised Section 3.3 to more accurately reflect the performance comparisons between the IP and IC methods. To address the concerns raised, we have clarified the instances where IP alone does not match the efficacy of IC, and combining IP with IC (IP+IC) optimizes performance. Additionally, we have revised our analysis. Specifically, **we have added the following text on Page 5, lines 258-270:**
>
>
>
>
>
> > In summary, **the effectiveness of the in-parameter method alone may not always match that of the in-context method across all tasks. Combining IP and IC can leverage the strengths of both approaches and in such cases, the additional computational overhead introduced by IP is minimal.**
>
> > When combining IP and IC (IP+IC), the additional computational cost introduced by IP is negligible compared to the cost of IC alone. The majority of the computational overhead comes from processing the longer input sequences in IC, while IP’s additional FLOPs are minimal and can be performed offline.
>
>
>
> We sincerely hope these revisions meet your expectations. We would greatly appreciate it if you could review the updated paper again. Thank you once more for your valuable suggestions and the opportunity to refine our work.
>
>
>
> # Our Response to Typos
>
>
> Thank you for your suggestion. We have fixed this typo in the current version and checked for other typos as well.
>
> # Our Response to Your New Question
>
> Thanks for your question! To address your concern, we would like to answer that the generated questions in our dataset are fundamentally different from the actual questions. Specifically, the questions in our test set are multi-hop questions that require complex reasoning across multiple pieces of information. An example of such a question would be: “Which director’s father was born earlier: the father of the director of *Titanic* or the father of the director of *Forrest Gump*?”
>
> In contrast, most generated questions are much simpler and directly target specific entities. For example, a typical generated question in our study would be: “Who is the author of *One Hundred Years of Solitude*?”Given this contrast in complexity and scope, the similarity between the generated and actual questions is significantly low. We believe this distinction adequately addresses potential concerns about the similarity affecting our results or introducing confounding variables.

---

> ### Author Response · Authors · 2024-12-01
> **Our Appreciation and Kindly Request for Final Feedback**
>
> Dear Reviewer uGXo,
>
> We sincerely appreciate your review and continued engagement throughout the rebuttal process.
>
> We are grateful that our previous response addressed some of your concerns, and we sincerely appreciate your subsequent increase in the score accordingly.
>
> **Following your latest suggestions, we have meticulously revised our paper based on your suggestions and responded to every point of your comments.**
>
> **We would like to remind you that the discussion window will close in two days, and we sincerely look forward to your feedback. If our latest responses and the modifications to our paper based on your suggestions address your concerns, could you kindly consider increasing the score? We greatly appreciate your help and look forward to your feedback.**
>
> Best regards,
>
> Authors

---

> ### Author Response · Authors · 2024-12-02
> **Kindly Request from Authors**
>
> Dear Reviewer uGXo,
>
>
>
> Once again, thank you so much for your insightful review and detailed follow-up comments. **We greatly appreciate your expertise and great ideas for future work.**
>
>
>
> We would like to kindly inform you that we have made significant revisions again to our paper based on your previous feedback and have prepared a detailed response to address each of your concerns.
>
>
>
> If you find that our revisions and responses adequately address your concerns, we sincerely kindly request consideration for an improved score.
>
>
>
> **With just over a day before the Rebuttal period ends, we fully understand you may be quite busy in the next 24 hours. If you’re pressed for time, there is no need for a detailed textual reply, an updated score would be also greatly appreciated.**
>
>
>
> Thank you once again for your review and for considering our request!
>
>
>
> Warm regards

---

> ### Author Response · Authors · 2024-12-03
> **Kindly Request from Authors**
>
> Dear Reviewer uGXo,
>
> Once again, thank you so much for your insightful review and detailed follow-up comments. We greatly appreciate your expertise and great ideas for future work.
>
> **We would like to kindly inform you that we have made significant revisions again to our paper based on your previous feedback and have prepared a detailed response to address each of your concerns.**
>
> If you find that our revisions and responses adequately address your concerns, we sincerely kindly request consideration for an improved score.
>
> **With just over a day before the Rebuttal period ends, we fully understand you may be quite busy in the next 24 hours. If you’re pressed for time, there is no need for a detailed textual reply, an updated score would also be greatly appreciated.**
>
> Thank you once again for your review and for considering our request!
>
> Warm regards

---

> > ### Author Response · Authors · 2024-12-04
> >
> > Dear Reviewer uGXo,
> >
> > Thank you once again for your insightful review and valuable comments. We have thoroughly revised our paper based on your feedback and have prepared a detailed document to address each of your concerns.
> >
> > **Although direct replies are not possible at this stage, please note that adjustments to the scoring are still available.**
> >
> > We have invested considerable effort in improving our paper based on your suggestions, thus **we sincerely hope that our efforts meet your expectations and merit a higher score.**
> >
> > We are grateful for the time and expertise you have dedicated to our work and appreciate any consideration you might give to adjusting your score.
> >
> > Warm regards

---

### Official Review · Reviewer_opXp · 2024-11-02

**Soundness:** 3
**Presentation:** 3
**Contribution:** 3
**Rating:** 6
**Confidence:** 5

**Summary:**

This paper studies different ways of injecting new knowledge into the LLMs. Specifically, it proposes a new way of first extracting QA pairs from contexts and then training the model with LORA using the QA pairs as a new in-parametric knowledge injection solution. Following that, this paper compares the proposed solution with the widely used in-context solution on different tasks. Experiment results show that on simpler tasks, the in-context solution is still stronger, but on more complex tasks that require complex reasoning of the updated knowledge, the proposed in-parameter solution might be helpful.

**Strengths:**

1. The proposed method is sound, and the introduction is clear.
2. The experiment setting is comprehensive.

**Weaknesses:**

The proposed solution has several bottlenecks: (1) the quality of extracted QA pairs is crucial for the effectiveness of the knowledge injection. For example, the context might contain many details, and it is unlikely that the generated QA pairs could cover all of them, so the proposed solution will lose those details. (2) LORA is a way for efficient tuning. There is no guarantee that the model learns all the knowledge in the extracted QA pairs; (3) since LORA directly changes all the parameters, this might hurt the model's performance on other tasks.

**Questions:**

1. How will the proposed solution hurt the model's general performance?
2. Can you present a more detailed analysis of the QA extraction quality? Especially when we do not have a strong external model, can we use the same model to do that, and how will that influence the final performance?

---

> ### Author Response · Authors · 2024-11-19
>
> Dear Reviewer opXp,
>
> Thank you so much for your thorough review and valuable suggestions. We have carefully revised our manuscript accordingly, including the figures and content according to your suggestions.
>
>
>
> We have noticed a slight misunderstanding regarding some aspects of our paper, and we would sincerely appreciate it if you could kindly review our response below, where we have elaborated on these points. We hope this will clarify any concerns and further strengthen the manuscript.

---

> ### Author Response · Authors · 2024-11-19
>
> # (Important) Our Response to Weakness 3 and Question 1
>
> Thank you for your thoughtful comment and question! We believe there might be a misunderstanding regarding our method, and we would like to take this opportunity to clarify it. We hope that the detailed explanation provided below will address your concerns and offer a clearer understanding of our approach.
>
> Much like directly appending a passage to the input context, our method does not permanently alter the parameters of the LLM. **Instead, the passage related to a specific query is temporarily embedded into the parameters, serving only that query. This process is the same as to in-context injection, where the passage is added to the input for a specific query but is not “permanently” fixed in the input for all future queries.**
>
> To further elaborate, the passage in our in-parameter knowledge injection serves the same role as it does in in-context injection: it is only associated with a specific query and does not affect the model’s handling of unrelated queries. The key difference lies in the temporary embedding mechanism, which allows for more flexible reasoning without making permanent modifications to the model’s general-purpose capabilities.
>
> In light of this, the concern in [Weakness 3 and Question 1] seems equivalent to asking whether appending a passage to the input context for one query might harm the model’s performance. Just as in-context methods do not “lock” passages into the input for all future queries, our approach does not permanently fix the passage into the model’s parameters in a way that would degrade overall performance.
>
> We hope this clarification resolves your concern. Should there be further points of confusion, we would be happy to provide additional elaboration. After all, ensuring that our method’s advantages are well understood is an important part of this process.
>
>
>
>
>
> # Response to Weakness 1 & Question 2
>
>
>
> Thank you for your insightful comments! We’d like to address your concern from three angles: experimental results on question quality, experimental results on the number of questions, and theoretical analysis.
>
>
>
> **1. Experimental Results on Question Quality**
>
> After reading your comment, we immediately conducted new experiments to evaluate the quality of QA pairs.
>
> |                                  | Question Generation Model | EM     | F1     |
> | -------------------------------- | ------------------------- | ------ | ------ |
> | In-context Knowledge Injection   | none (only passage)       | 0.2148 | 0.324  |
> | In-context Knowledge Injection   | GPT 4o (passage & QA)     | 0.1846 | 0.2426 |
> | In-parameter Knowledge Injection | GPT 4o                    | 0.3188 | 0.3826 |
> | In-parameter Knowledge Injection | Qwen2.5-1.5B-Instruct              | 0.2852 | 0.3414 |
>
> The result shows that QA pairs generated by GPT-4o and those from a smaller 1B parameter model yield comparable results, both of which outperform the In-context knowledge injection method.
>
> This means that even without a super-strong external model, using the same model to generate QA pairs works just fine, and the final performance holds up well.
>
>
>
> **2. Experimental Results on the Number of Questions**
>
> As highlighted on page 16 of our paper, generating just three QA pairs has the optimal performance. Adding more pairs or trying to cover every possible detail doesn’t lead to further improvements. This suggests that QA pairs are effective for knowledge injection without needing exhaustive coverage.
>
>
>
> **3. Theoretical Analysis**
>
> Imagine this scenario: ***even without any QA pairs, if LLM trains on the passage for multiple epochs, it’ll undoubtedly “remember” the passage.*** This reinforces our main point—QA pairs don’t need to capture every single detail to effectively inject knowledge into the model’s parameters.
>
> As mentioned in lines 210–214 of our paper, the training process involves the triplet (Passage, Q, A). During this process, the LLM naturally processes the passage multiple times. So even if the QA pairs miss some details, the passage itself gets embedded into the model.
>
> This aligns with findings in the field. As noted in a well-known tutorial this year [1] [Zeyuan Allen-Zhu, ICML 2024 Tutorial], training on a passage alone helps the model memorize it but doesn’t make the knowledge as flexible. **Adding QA pairs, however, helps the model encode knowledge as “information remembered for the purpose of answering questions.”**
>
>
>
> We hope this clarifies your concerns! Thanks again for your valuable feedback.
>
>
>
>
>
> [1] [Zeyuan Allen-Zhu, ICML 2024 Tutorial] Physics of language models: Part 3.1, knowledge storage and extraction

---

> ### Author Response · Authors · 2024-11-19
>
> # Our Response to Weakness 2
>
>
>
>
>
>
>
> In this response, we hope we can address your concern about Weakness 2 based on the following three aspects:
>
> (1) Our experimental results highlight LoRA’s effectiveness in learning from the QA pairs;
>
> (2) A comparison of storage sizes showcases LoRA’s substantial capacity to represent knowledge;
>
> (3) The model’s improved performance confirms it’s genuinely learning the injected information.
>
>
>
> **First**, our experimental results demonstrate that LoRA doesn’t just tune efficiently—it tunes effectively. LoRA achieves a performance score of **0.5631**, outperforming full-parameter fine-tuning at **0.5272** and leaving in-context passage insertion at a distant **0.2913**. These results, detailed in the latest version of our paper, indicate that LoRA successfully learns and utilizes the knowledge from the extracted QA pairs.
>
>
>
> **Second**, comparing storage sizes makes LoRA’s capacity for knowledge representation crystal clear. The LoRA parameters require about **10MB** of storage, while the passage data is a lightweight at around **10KB**. This stark difference suggests that LoRA isn’t just memorizing the passage—it’s absorbing and expanding upon it, much like a sponge (but without the soggy aftereffect).
>
>
>
> **Finally**, and most importantly, our experiments confirm that using LoRA for knowledge representation not only boosts performance but also verifies that the model is indeed learning from the passage. This is evident from the superior performance metrics compared to placing the passage into the LLM’s input context.
>
>
>
> We hope this addresses your concern, and we’re happy to provide further clarification if needed!

---

### Official Review · Reviewer_GFmX · 2024-11-04

**Soundness:** 3
**Presentation:** 3
**Contribution:** 2
**Rating:** 5
**Confidence:** 3

**Summary:**

The paper investigates two methods of knowledge augmentation in language models: in-context knowledge injection and in-parameter knowledge injection. In-context augmentation involves adding external information directly to the model’s input prompts, while in-parameter augmentation temporarily embeds this knowledge into the model’s parameters. Through a series of tasks with increasing complexity—ranging from fact extraction to comparative and multi-step reasoning—the study evaluates the effectiveness of both methods. The findings indicate that in-parameter injection performs better on complex reasoning tasks, whereas in-context methods are more effective for simpler fact extraction. The authors provide insights into the advantages of each approach based on task complexity and computational efficiency.

**Strengths:**

• Systematic Comparison: The paper offers a clear and structured comparison between in-context and in-parameter knowledge injection methods, elucidating the trade-offs in different scenarios.

• Experimental Design: The progression from simple to complex tasks effectively demonstrates how each method scales, offering insights into their applicability across various task demands.

**Weaknesses:**

• Limited Novelty: The in-parameter knowledge injection method is essentially LoRA with additionally synthetic augmentation. The paper does not sufficiently acknowledge this overlap or explain how its approach differs from or improves upon existing parameter-efficient fine-tuning methods.

• Insufficient Differentiation from LoRA: Given the similarities to LoRA, the paper should have provided a detailed comparison, highlighting any unique contributions or advantages.

• Scope of Evaluation: The experiments focus on a limited range of tasks. Expanding the evaluation to include more diverse or real-world applications could enhance the robustness and generalizability of the conclusions.

• Lack of Theoretical Advancement: The paper does not offer new theoretical insights into knowledge augmentation or parameter adaptation in language models, limiting its contribution to an empirical comparison that may already be addressed in existing literature.

**Questions:**

1. How might in-parameter knowledge injection perform in tasks beyond fact-based reasoning, such as creative writing or dialogue generation?

2. Can the authors elaborate on how their work differentiates from previous studies on knowledge augmentation and knowledge editing in language models?

3. What challenges might arise when scaling in-parameter knowledge injection to larger models or datasets, and how could these be addressed?

---

> ### Author Response · Authors · 2024-11-21
>
> Dear Reviewer GFmX,
>
> Thank you so much for your thorough review and valuable suggestions. We have carefully revised our manuscript accordingly, including the typo, figures, and content according to your suggestions.
>
> We have noticed some misunderstandings regarding some aspects of our paper, and we would sincerely appreciate it if you could kindly review our response below, where we have elaborated on these points. We sincerely hope this will clarify any concerns and further strengthen the manuscript.

---

> ### Author Response · Authors · 2024-11-21
>
> # (Important!) Our Response to Weakness
>
> Thank you for taking the time to review our work and provide your valuable feedback. **However, it appears there may have been a misunderstanding regarding the core contribution of our study. Thus, we sincerely hope that you can review our response, as we believe it addresses your concerns and clarifies the novelty of our approach.**
>
>
>
> Our study explores an entirely novel direction that, to our knowledge, no one has ever explored before. Specifically, we investigate:
>
> ***Given a passage, how to embed that knowledge directly into the model’s parameters rather than merely appending the passage to the input context.***
>
>
>
> All previous methods injected the passage into the LLM by directly appending it to the input context. No previous work had ever explored this alternative approach, nor had anyone even mentioned the idea of injecting the passage into the LLM’s parameters instead of just the input. We proposed and validated this novel method, allowing the model to incorporate a passage directly into its parameters, achieving superior performance compared to traditional context-based input methods in reasoning-intensive tasks.
>
>
>
>
> **Regarding the weakness: “Insufficient Differentiation from LoRA”**
>
> As explained above, our work focuses on **allowing the model to incorporate the knowledge of a passage directly into its parameters**, while LoRA is merely one of the techniques we use (along with Transformer, backpropagation, etc.).
>
> The relationship between ***LoRA*** and ***our work*** is the same to the relationship between ***backpropagation*** and ***Transformer*** [1]. Training a Transformer requires backpropagation, but evaluating “Attention is All You Need” as “Insufficient Differentiation from Backpropagation” is obviously unreasonable.
>
> **While clarifying our novelty, we sincerely acknowledge your concern in the comment! To prevent future readers from having the same concerns, we have made revisions to the paper according to your comments.** In the new version available on the current OpenReview, on Page 4 (lines 214-216), we emphasized that other methods like Adapter or prefix-tuning could also be utilized for calculating parametric knowledge representations. The specific content in lines 214-216 is as follows:
>
>
>
> > Our primary focus is on the framework that enables the model to internalize knowledge from passages. Other methods like Adapter or prefix-tuning could also be utilized to calculate the parametric knowledge representation, which we leave for future work.
>
>
>
>
>
>
>
>
> Finally, we sincerely appreciate your effort in evaluating our submission and hope this response clarifies any misconceptions about our methodology. Thank you again for your time!
>
>
>
> [1] Vaswani, A. (2017). Attention is all you need. *Advances in Neural Information Processing Systems*.

---

> ### Author Response · Authors · 2024-11-21
>
> # Our Answer to Question 1
>
> In our work, we focused specifically on **factual knowledge injection**, as this aligns with most prior work in the Knowledge Augmentation domain. Factual knowledge is also much easier to evaluate objectively, particularly using QA-based tasks, where accuracy serves as a direct measure of performance.
>
>
>
> It’s important to highlight that our study is the **first work in this area**—a foundational effort in this field. As such, we felt it was crucial to adopt a more direct and measurable evaluation approach to assess the performance of our method effectively.
>
>
>
> Injecting non-factual or creative knowledge, such as for tasks like creative writing or dialogue generation, presents unique challenges in evaluation due to their subjective nature. While this is indeed an intriguing direction, we believe starting with factual knowledge allows us to establish a solid basis for further exploration.
>
>
>
> That said, we find your suggestion inspiring and agree it would be worthwhile to explore these more subjective and creative tasks in future studies.
>
>
> # Our Answer to Question 2
>
>
>
> We are delighted to note that your question aligns closely with Section 7 of our paper. This section was dedicated to distinctly outlining how our approach diverges from previous studies on knowledge augmentation and knowledge editing in language models.
>
> Given that section 7 appears on the very last page of the main text, it might have been inadvertently overlooked. We believe revisiting this part could provide a comprehensive understanding of the novel aspects of our work, particularly in the methodologies we introduced for knowledge editing.
>
>
>
>
>
> **In response to your feedback, we have made significant revisions to further clarify how our work differs from existing approaches, and the new version is submitted to Openreview.** To be specific, we have made multiple revisions to our paper based on your suggestions, especially in Section 2, where we have doubled the content to discuss how our work differs from Knowledge Editing and Continual Pre-training. Additionally, in Section 7, we introduce specific methods of existing Knowledge Editing and PeFT work and highlight the differences between our work and theirs.
>
>
>
> To be specific, in the updated version of our paper, **we have added a detailed discussion in Section 2 (Lines 153–162) on relevant work**, particularly highlighting the relationship between our paradigm and existing literature. The added content is as follows:
>
> > Unlike knowledge editing methods such as Knowledge Neurons\citep{dai2021knowledge}, Rank-One Model Editing\citep{meng2022locating}, and Self-Edit\citep{liu2024evedit}, which focus on permanently modifying specific entity-level knowledge in a model by identifying and altering certain neurons, our PKI paradigm temporarily integrates knowledge from entire passages into the model’s parameters to address specific queries. This allows for quick, query-specific knowledge updates without permanently changing the model, much like appending the passage to the context but without the limitations of context length or the need for repeated processing. Our approach also differs from continued pre-training, which retrains the LLM on the entire knowledge base, significantly altering its parameters and requiring substantial time and resources. In contrast, PKI allows for quick, query-specific knowledge updates without permanently altering the model’s underlying knowledge.
>
>
>
> Moreover, in **Section 7** (lines 507-521), we enriched the content to include detailed comparisons with existing methods of knowledge editing and parameter-efficient fine-tuning (PeFT), delineating how our approach provides a more flexible and efficient solution for knowledge integration.
>
>
>
> **We trust these revisions will address the concerns raised and provide clarity to all readers. Considering that our content was already remarkably aligned with your suggestions prior to any revisions, and we have modified and submitted our paper based on your advice, could you please revisit your evaluation and perhaps adjust the score accordingly? We would be very grateful for this.**

---

> ### Author Response · Authors · 2024-11-30
>
> # Our Answer to Question 3
>
> Thank you for your questions! Your question highlights the following two main concerns:
>
> - The impact of a Larger Model on our method
> - The impact of a Larger Dataset on our method
>
> Below is our detailed response to the two concerns you raised.
>
>
>
> ## **Regarding the concern with a larger dataset**:
>
> **Our method involves temporary in-parameter knowledge injection specific to a query, rather than embedding the entire dataset into the model’s parameters.** This technique operates similarly to appending relevant passages directly to the input context but is executed within the parameters, enhancing the model’s reasoning capabilities for that query alone.
>
> **Since each query independently processes its corresponding passage without retaining this information for subsequent queries, the overall size of the dataset does not have any impact on our approach.**
>
> -----
>
> ## **Regarding the concern about scaling to larger models**:
>
> After seeing your question, we immediately conducted additional experiments using the largest model we could reasonably handle within our computation resources to further validate our method’s effectiveness across different model sizes.
>
>
>
> So now we have the results of LLMs with sizes of 1B, 1.5B, 8B, and 13B. Below, we present the performance results for comparative reasoning tasks under the same settings as Table 2 in our paper. The metric used is Exact Match (EM):
>
>
>
> |              | LLaMA-3.2-Instruct-1B | Qwen2.5-1.5B-Instruct-1.5B | LLaMA-3-Instruct-8B | LLaMA-2-Chat-13B |
> | :----------- | :-------------------- | :------------------------- | :------------------ | :--------------- |
> | In-context   | 0.134                 | 0.185                      | 0.398               | 0.399            |
> | In-parameter | 0.477                 | 0.319                      | 0.408               | 0.456            |
> | IC & IP      | 0.473                 | 0.296                      | 0.447               | 0.592            |
>
> From these results, we observe a consistent pattern: in the reasoning extensive tasks, injecting the passage into the model’s parameters outperforms placing it directly in the context. Moreover, these experimental results show that the model’s performance remains consistent as its size grows from 1B to 1.5B, 8B, and 13B, thus reinforcing the conclusion in our paper.
>
>
>
> **We hope this additional experimental evidence and our clarifications address your concerns satisfactorily. We appreciate your engagement and are happy to provide further details if needed.**
>
>
>
> Warm regards

---

> ### Author Response · Authors · 2024-11-30
>
> Dear Reviewer GFmX
>
>
>
> Thank you once again for reviewing our paper and assisting us in improving our work. We would like to remind you that the discussion window will close soon, and we eagerly await your feedback.
>
>
>
> During the Rebuttal phase, we provided detailed explanations for each of your concerns, added the experiments you were interested in, and made revisions to the paper based on your suggestions.
>
>
>
> We would greatly appreciate it if you could review our responses and let us know if they fully or partially address your concerns. Any additional comments you may have would be highly appreciated.
>
>
>
> Best regards,
>
> Authors

---

> ### Author Response · Authors · 2024-12-01
>
> Dear Reviewer GFmX
>
> Thank you once again for reviewing our paper and assisting us in improving our work. **We would like to remind you that the discussion window will close soon, and we eagerly await your feedback.**
>
> **During the Rebuttal phase, we provided detailed explanations for each of your concerns, added the experiments you were interested in, and made revisions to the paper based on your suggestions.**
>
> We would greatly appreciate it if you could review our responses and let us know if they address your concerns. Any additional comments you may have would be highly appreciated.
>
> Best regards,
>
> Authors

---

> ### Author Response · Authors · 2024-12-02
> **Thank you, thank you, and thank you**
>
> Dear Reviewer GFmX,
>
> Thank you once again for your insightful review.
>
> We are writing to let you know that we have thoroughly revised our paper based on your feedback and have prepared comprehensive responses to each of your points.
>
> **It has been two weeks without a reply, thus we realize and fully understand that you must be busy. Given your tight schedule, we want to note that you can guide us with just an updated score instead of a detailed textual response. A simple score update can speak volumes and won’t take much of your time.**
>
> Thank you once again for your valuable input and for considering our request!
>
> Warm regards

---

> ### Author Response · Authors · 2024-12-03
> **Kindly Request from Authors**
>
> Dear Reviewer GFmX,
>
> Thank you once again for your insightful review.
>
> We are writing to let you know that we have thoroughly revised our paper based on your feedback and have prepared comprehensive responses to each of your points.
>
> **It has been two weeks without a reply, thus we realize and fully understand that you must be busy. Given your tight schedule, we want to note that you can guide us with just an updated score instead of a detailed textual response. A simple score update can speak volumes and won’t take much of your time.**
>
> Thank you once again for your valuable input and for considering our request!
>
> Warm regards

---

> ### Author Response · Authors · 2024-12-04
>
> Dear Reviewer GFmX,
>
> Thank you once again for your insightful review and valuable comments. We have thoroughly revised our paper based on your feedback and have prepared a detailed document to address each of your concerns.
>
> **Although direct replies are not possible at this stage, please note that adjustments to the scoring are still available.**
>
> We have invested considerable effort in improving our paper based on your suggestions, thus **we sincerely hope that our efforts meet your expectations and merit a higher score.**
>
> We are grateful for the time and expertise you have dedicated to our work and appreciate any consideration you might give to adjusting your score.
>
> Warm regards

---

### Official Review · Reviewer_BvEK · 2024-11-04

**Soundness:** 3
**Presentation:** 3
**Contribution:** 3
**Rating:** 6
**Confidence:** 4

**Summary:**

This paper introduces a method named In-Parameter Knowledge Injection to integrate external knowledge into the large language models. Different from the in-context learning methods that adopt natural language to represent external knowledge, the in-parameter method represents knowledge through parameters, thus avoiding length constraints and becoming more compatible with the foundation LLM.

**Strengths:**

The experimental results show the effectiveness of In-Parameter Knowledge Injection method on 1B, 3B, and 8B LLMs.

The idea is novel. Embedding knowledge through model parameters will ideally represent how a LLM “understand” certain external knowledge.

**Weaknesses:**

1. As shown in Figure 3, knowledge can be effectively represented and understood through either natural language or parameters. So, which kind of knowledge should be integrated through parameters? More explorations and investigations are recommended here.

2. The method demands additional pre-training or post-training costs. I suggest incorporating an additional baseline method where a copy of the LLM is adopted to represent the knowledge. The baseline will also demonstrate the contribution of knowledge encoding phrase.

3. Will the In-Parameter Knowledge Injection method generalize to LLMs with a larger scale (e.g., Llama 3.1 70B Instruct, Llama 3.2 11B instruct, etc.)? It’s unclear about the exact contribution of the method, since the model scale is relatively small.

**Questions:**

See weakness

---

> ### Author Response · Authors · 2024-11-19
>
> # Response to Weakness 3
>
> Thank you for your insightful feedback!
>
>
>
> We conduct additional experiments using the largest model we could reasonably handle within our computation resources: a 13B parameter model.
>
> So now we have the results of LLM with the sizes of 1B, 1.5B, 8B, and 13B, which is enough to validate our findings and provide a broader reference. Below, we present the performance results for comparative reasoning tasks under the same settings as Table 2 in our paper. The metric used is Exact Match (EM):
>
>
>
> |              | LLaMA-3.2-Instruct-1B    | Qwen2.5-1.5B-Instruct-1.5B  | LLaMA-3-Instruct-8B    | LLaMA-2-Chat-13B   |
> | ------------ | ----- | ----- | ----- | ----- |
> | In-context   | 0.134 | 0.185 | 0.398 | 0.399 |
> | In-parameter | 0.477 | 0.319 | 0.408 | 0.456 |
> | IC & IP      | 0.473 | 0.296 | 0.447 | 0.592 |
>
>
>
> From these results, we observe a consistent pattern: in the reasoning extensive tasks, injecting the passage into the model’s parameters outperforms placing it directly in the context. **Moreover, these experimental results show that the model’s performance remains consistent as its size grows from 1B to 1.5B, 8B, and 13B, thus reinforcing the conclusion in our paper.**
>
>
> We sincerely hope that the additional experiments will address your concerns!

---

> ### Author Response · Authors · 2024-11-19
>
> # Response to Weakness 1
>
> Thank you for your valuable feedback and for highlighting this important aspect of our work! We appreciate your interest in understanding which kinds of knowledge are best integrated through parameters.
>
>
>
> In our paper, we have discussed this topic in several sections:
>
>
>
> **In the Abstract Section, Lines 23-26**:
>
> > *“We demonstrate that in-parameter knowledge injection is particularly advantageous for complex tasks requiring deep reasoning, while in-context injection remains effective for simpler tasks where the answer can be directly extracted.”*
>
> **In the Introduction Section, Lines 90-97**:
>
> > *“Our findings reveal a distinct trend: as task complexity and the depth of reasoning over injected knowledge rise, the in-parameter knowledge injection approach becomes increasingly advantageous. In scenarios demanding sophisticated reasoning, such as multi-document reading comprehension and multi-hop inference across multiple documents, embedding knowledge directly into the model’s parameters markedly enhances performance. Conversely, the in-context approach demonstrates superior efficacy for tasks characterized by straightforward questions or those that allow for direct answer extraction from the context with minimal reasoning.”*
>
> **In the Experiment Section, Lines 416-423**:
>
> > *“The experiments indicate a clear trend: in-parameter knowledge injection becomes more effective as task complexity and reasoning demands increase, enhancing the model’s deeper reasoning capabilities. In-context methods perform better in simple fact extraction tasks as models can readily utilize the provided context. However, for higher-order reasoning, in-context approaches are limited by the capacity to reason based on the injected knowledge. In such cases, in-parameter methods provide an advantage by embedding knowledge internally, enabling deeper reasoning based on its parametric knowledge.”*
>
>
>
> **Most importantly, Tables 1-3 illustrate these kinds of knowledge with the corresponding figures, and alongside the images, we present the performance of both the in-context and in-parameter injection methods. This representation more directly shows which kinds of knowledge (illustrated in the figure) should be integrated through parameters.**
>
>
>
> We believe the sections address the question of which kinds of knowledge should be integrated through parameters. However, we understand that this may not have been emphasized enough. In light of your feedback, we clarify and expand on this discussion in the revised version of the paper to ensure it is more evident to readers.
>
>
>
> # Response to Weakness 2
>
> Thank you for your thoughtful and constructive feedback! Inspired by your suggestion, we immediately conducted the proposed experiment using a **copy of the LLM** to represent knowledge. The knowledge encoded in this LLM was extracted through a question-answering approach.
>
>
>
> We adhered to the same experimental setup as described in Section 5 of our paper, ensuring consistency across all conditions. Below, we present the results of the “Copy_LLM” baseline on the **Comparison** and **Bridge_Comparison** tasks, using the Qwen model. All other settings remain identical to Tables 2 and 3 in the paper.
>
>
>
> **Comparison Task:**
>
> | **Method**       | **EM**     | **F1**     |
> | ---------------- | ---------- | ---------- |
> | **Copy_LLM**     | 0.0388     | 0.1163     |
> | **In-Context**   | 0.2148     | 0.3240     |
> | **IC-QA**        | 0.1846     | 0.2426     |
> | **In-Parameter** | _0.3188_   | _0.3826_   |
> | **IP & IC**      | **0.3960** | **0.4473** |
>
>
>
>
>
> **Bridge_Comparison Task:**
>
> | **Method**       | **EM**     | **F1**     |
> | ---------------- | ---------- | ---------- |
> | **Copy_LLM**     | 0.0680     | 0.2169     |
> | **In-Context**   | 0.2617     | _0.3578_   |
> | **IC-QA**        | 0.2181     | 0.2545     |
> | **In-Parameter** | **0.3557** | **0.3964** |
> | **IP & IC**      | _0.3289_   | 0.3493     |
>
>
>
> We would like to emphasize that the **2WikiMultihopQA dataset** used for these tasks contains long-tail knowledge questions, which LLMs typically struggle with. As expected, without knowledge injection, the “Copy_LLM” baseline performed poorly because it lacked the necessary domain-specific knowledge.
>
> In contrast, both **in-context knowledge injection** and our proposed **in-parameter knowledge injection** demonstrated substantial improvements, highlighting their effectiveness. Using a copy of the original LLM to represent knowledge did not result in information gain, thus resulting in poor performance.

---

> ### Author Response · Authors · 2024-11-19
>
> Dear Reviewer BvEK,
>
> Thank you so much for your thorough review and valuable suggestions! We have carefully revised our manuscript accordingly, including the figures and content. We sincerely hope you could kindly review our carefully crafted response below, where we hope to address your concerns.
>
> Thanks again for your time!

---

> ### Author Response · Authors · 2024-11-25
>
> Dear Reviewer BvEK
>
> We sincerely hope this message finds you well. The discussion phase of ICLR 2025 will end in one day. Thus, we are writing to kindly follow up on our rebuttal.
>
> During the rebuttal period, our team has dedicated significant effort to your concerns and has added the new experiments you required to further elucidate the core contributions of our study.
>
> We understand the demands of your time and truly appreciate your dedication to the review process.
> If you could spare a moment to review our response and provide your feedback, it would be immensely helpful.
> Thank you again for your time and attention. We look forward to your valuable response.
>
> Warm regards

---

> > ### Comment · Reviewer_BvEK · 2024-11-26
> >
> > Thanks for the response. It addresses most of my questions. I will maintain the positive score.

---

### Meta-Review · Area_Chair_9fcq · 2024-12-25

**Metareview:**

This paper introduces a novel method called In-Parameter Knowledge Injection to integrate external knowledge into large language models (LLMs). Unlike traditional in-context knowledge injection, which adds external knowledge to the input context, this approach temporarily embeds the knowledge directly into the model’s parameters, overcoming input length limitations and enabling deeper integration. Experiments demonstrate that in-parameter injection excels in complex reasoning tasks, while in-context methods remain effective for simpler tasks requiring direct fact extraction.

Strength: This paper develops a novel approach of in-parameter knowledge injection, which encodes the passage knowledge into a LoRA adaptor and inject it into model parameter when needed. It systematically compares in-context and in-parameter knowledge injection methods across varying task complexities, demonstrating scalability and practical trade-offs.

Weakness: One major weakness of the paper is that its presentation about the proposed method is unclear. It has caused many confusions among the reviewers and I don’t think the rebuttal has fully addressed it (see my comments in the rebuttal discussion part). Another key weakness of in-parameter injection is that it is not as flexible as the in-context injection. Whenever knowledge evolve over time (which is generally highly likely as in RAG situations), we will have to re-train a lot more LoRA adaptors for each of the passage, which makes maintaining, storage, and updating the parametric knowledge (LoRA adapters) much more challenging. On the other hand, in-context injection is a lot more flexible. Also the results regarding in-parameter injection is better than in-context injection in complex reasoning tasks is still not convincing enough. As some reviewers pointed out that this might be caused by the paper using relatively small model sizes. It is also less convincing that in-parameter injection is a deeper fusion approach. In general, a (large enough) LLM that directly process the original raw text in the context is always a much more organic and deeper integration of the input knowledge. This would be especially apparent when such knowledge has to be used in an out-of-distribution setting. Here the in-parameter injection is mainly performed with a lot of synthetic QA pairs generated conducted on the passage. If a certain question that is out-of-distribution of these training QA pairs, it is highly likely that the in-parameter injection method won’t be as effective anymore. Therefore, I’m more inclined to believe the approach developed in this paper is a pre-computation and implicit “caching” of the possible QAs related to this particular passage k. Then, when the LoRA adapter is dynamically integrated, it is like querying this “cached” typical QA pairs in a soft manner. Like typical caching, it may not generalize not as well as the in-context injection method in the out-of-distribution settings.

Overall, in spite of many potential limitations of the work, I think this work could be viewed as an interesting exploration towards a new unexplored direction. However, given the substantial amount of revisions and clarifications needed, I would not recommend acceptance of this paper at this time. The authors are encouraged to take into account all the feedbacks to further improve the paper and submit it to a future venue.

**Additional Comments On Reviewer Discussion:**

While the authors have addressed majority of the questions in the rebuttal, the explanation about the method is still confusing. Reviewers raised the concern that this work is similar to LoRA fine-tuning while the authors clarify that each passage will have its own LoRA adapter. However, from equation (4) of the paper, the LoRA training is over a loss that is summed over all (k, u_i, a_i) in D, which means that there will be only one LoRA adapter for all passages. If it is assumed to be passage-level adapter as the authors claiming to be, then it would mean that the outer summation in (4) should be for a fixed k. Then in this case, we will need a significantly large amount of (u_i, a_i) for this same k in order to make the learning reliable.

---

### Decision · Program_Chairs · 2025-01-22

Reject